# Fail-Safe Adversarial Generative Imitation Learning

**Philipp Geiger**                                                     *Philipp.W.Geiger@de.bosch.com*
*Bosch Center for Artificial Intelligence*
*Renningen, Germany*

**Christoph-Nikolas Straehle**                           *Christoph-Nikolas.Straehle@de.bosch.com*
*Bosch Center for Artificial Intelligence*
*Renningen, Germany*

**Reviewed on OpenReview:** *https://openreview.net/forum?id=e4Bb0b3QgJ*

## Abstract

For flexible yet safe imitation learning (IL), we propose theory and a modular method, with a safety layer that enables a closed-form probability density/gradient of the safe generative continuous policy, end-to-end generative adversarial training, and worst-case safety guarantees. The safety layer maps all actions into a set of safe actions, and uses the change-of-variables formula plus additivity of measures for the density. The set of safe actions is inferred by first checking safety of a finite sample of actions via adversarial reachability analysis of fallback maneuvers, and then concluding on the safety of these actions' neighborhoods using, e.g., Lipschitz continuity. We provide theoretical analysis showing the robustness advantage of using the safety layer already during training (imitation error linear in the horizon) compared to only using it at test time (up to quadratic error). In an experiment on real-world driver interaction data, we empirically demonstrate tractability, safety and imitation performance of our approach.

## 1 Introduction and Related Work

For several problems at the current forefront of agent learning algorithms, such as decision making of robots or automated vehicles, or modeling/simulation of realistic agents, imitation learning (IL) is gaining momentum as a method (Suo et al., 2021; Igl et al., 2022; Bhattacharyya et al., 2019; Bansal et al., 2018; Xu et al., 2020; Zeng et al., 2019; Cao et al., 2020). But *safety* and *robustness* of IL for such tasks remain a challenge.

**(Generative) IL**    The basic idea of imitation learning is as follows: we are given recordings of the sequential behavior of some *demonstrator* agent, and then we train the *imitator* agent algorithm on this data to make it behave similarly to the demonstrator (Osa et al., 2018). One powerful recent IL method is *generative adversarial imitation learning (GAIL)* (Ho and Ermon, 2016; Song et al., 2018). GAIL's idea is, on the one hand, to use policy gradient methods borrowed from reinforcement learning (RL; including, implicitly, a form of *planning*) to generate an imitator policy that matches the demonstrator's behavior over whole trajectories. And on the other hand, to measure the "matching", GAIL uses a *discriminator* as in *generative adversarial networks (GANs)* that seeks to distinguish between demonstrator and imitator distributions. While the neural net and GAN aspects help the flexibility of the method, the RL aspects stir against compounding of errors over rollout horizons, which other IL methods like *behavior cloning (BC)*, which do not have such planning aspects, suffer from. One line of research generalizes classic Gaussian policies to more flexible (incl. multi-modal) conditional *normalizing flows* as imitator policies (Ward et al., 2019; Ma et al., 2020) which nonetheless give exact densities/gradients for training via the *change-of-variables formula*.

**IL's safety/robustness challenges**    While these are substantial advances in terms of *learning flexibility/capacity*, it remains a challenge to make the IL methods (1) guaranteeably *safe*, especially for multi-agent

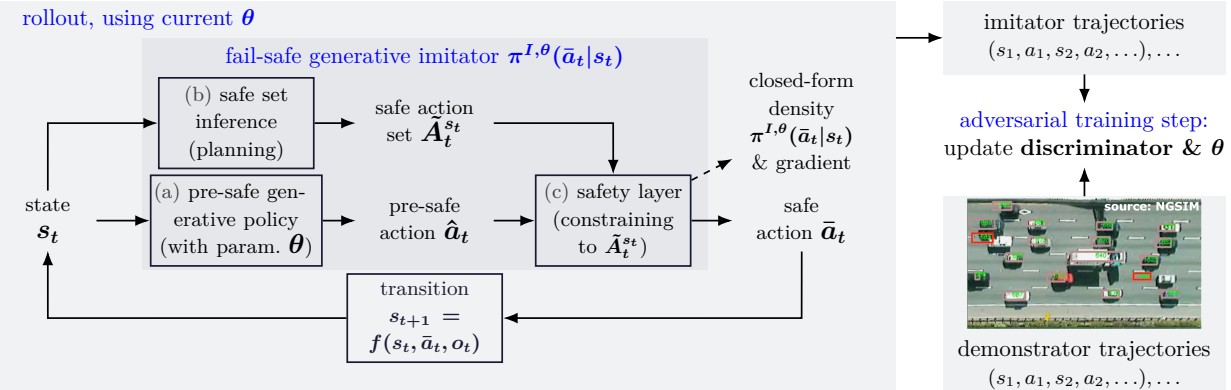

Figure 1: Outline of our theoretically grounded method *FAGIL*: It consists of the *fail-safe generative imitator policy* (inner box on l.h.s.), that takes current state $s_t$ as input, and then it (a) lets a standard pre-safe generative policy (e.g., Gauss or normalizing flow) generate a pre-safe action $\hat{a}_t$, (b) *infers a provably safe set $\tilde{A}_t^{s_t}$ of actions*, and then (c) uses our *safety layer* to constrain the pre-safe action $\hat{a}_t$ into $\tilde{A}_t^{s_t}$, yielding safe action $\bar{a}_t$. It has *end-to-end closed-form density/gradient*, so we train it end-to-end via, e.g., GAIL (r.h.s.). While our approach is for general safe IL tasks, we take driver imitation as running example/experiments.

interactions with humans, and (2) *robust*, in the sense of certain generalizations "outside the training distribution", e.g., longer simulations than training trajectories. Both is non-trivial, since safety and long-term behavior can heavily be affected already by small learning errors (similar as the *compounding error* of BC).

**Safe control/RL ideas we build on**   To address this, we build on several ideas from safe control/RL: (1) In reachability analysis/games (Fridovich-Keil and Tomlin, 2020; Bansal et al., 2017), the safety problem is formulated as finding controllers that provably keep a dynamical system within a pre-defined set of safe/collision-free states over some horizon, possibly under adversarial perturbations/other agents.  (2) Any learning-based policy can be made safe by, at each stage, if it *fails* to produce a safe action, *enforcing* a safe action given by some sub-optimal but safe fallback controller (given there is such a controller) (Wabersich and Zeilinger, 2018). (3) Often it is enough to search over a reasonably small set of candidates of fallback controllers, such as emergency brakes and simple evasive maneuvers in autonomous driving (Pek and Althoff, 2020). And (4) a simple way to enforce a policy's output to be safe is by composing it with a *safety layer* (as final layer) that maps into the safe set, i.e., constrains the action to the safe set, and *differentiable* such safety layers can even be used during *training* (Donti et al., 2021a; Dalal et al., 2018).

**Safe IL, theoretical guarantees, and their limitations**   While incorporating safety into *RL* has received significant attention in recent years (Wabersich and Zeilinger, 2018; Berkenkamp et al., 2017; Alshiekh et al., 2018; Chow et al., 2019; Thananjeyan et al., 2021), safety (in the above sense) in *IL*, and its *theoretical analysis*, have received comparably little attention. The existing work on such safe IL can roughly be classified based on whether safety is incorporated via *reward augmentation* or via *constraints/safety layers encoded into the policy* (there also exists a slightly different sense of safe IL, with methods that control the probabilistic risk of high imitation costs (Javed et al., 2021; Brown et al., 2020; Lacotte et al., 2019), but they do not consider safety constraints *separate* from the imitation costs, as we do; for additional, broader related work, see Appendix D.4). *In reward augmentation*, loss terms are added to the imitation loss that penalizes undesired state-action pairs (e.g., collisions) (Bhattacharyya et al., 2019; 2020; Bansal et al., 2018; Suo et al., 2021; Zeng et al., 2020; Cheng et al., 2022). *Regarding hard constrains/safety layers* (Tu et al., 2022; Havens and Hu, 2021), one line of research (Chen et al., 2019) is deep IL with safety layers, but *only during test time*.  There is little work that uses safety layers during *training and test time*, with the exception of (Yin et al., 2021) which gives guarantees but is not generative.

**Main contributions and paper structure**   In this sense we are not aware of general approaches for the problem of safe generative IL that are (1) safe/robust with theoretical guarantees *and* (2) end-to-end

trainable. In this paper, we contribute theory (Sec. 3), method (Fig. 1, Sec. 4) and experiments (Sec. 5) as one step towards understanding and addressing these crucial gaps:[1][2]

- We propose a simple yet flexible type of differentiable *safety layer* that maps a given "pre-safe" action into a given safe action set (i.e., constrains the action). It allows to have a *closed-form probability density/gradient* (Prop. 3; by a non-injective "piecewise" change of variables) of the overall policy, in contrast to common existing differentiable safety layers (Donti et al., 2021a; Dalal et al., 2018) without such analytic density.

- We contribute two *sample-based safe set inference approaches*, which, for a given state, output a provably safe set of actions – in spite of just checking safety of a *finite* sample of actions, using Lipschitz continuity/convexity (Prop. 1 and 2). This advances sample-based approaches (Gillula et al., 2014) to overcome limitations of exact but restrictive, e.g., linear dynamics, approaches (Rungger and Tabuada, 2017).

- A general question is to what extent it helps to use the *safety layer already during training (as we do)*, compared to just concatenating it, at *test* time, to an unsafely trained policy, given the latter may computationally be much easier. We theoretically quantify the imitation performance advantage of the former over the latter: essentially, the former imitation error scales *linearly* in the rollout horizon, the latter up to *quadratically* (Sec. 3.3, rem. 1, and thm. 1) – reminiscent of BC. The intuition: only the former method learns how to properly deal (plan) with the safety layer, while in the latter case, the safety layer may *lead to unvisited states at test time from which we did not learn to recover.* (Generally, proofs are in Appendix A.)

- Combining these results, we propose the method *fail-safe generative adversarial imitation learner (FAGIL;* Sec. 4). – It is sketched in Fig. 1.

- In experiments on real-world highway data with multiple interacting driver agents (which also serves as running example throughout the paper), we empirically show tractability and safety of our method, and show that its imitation/prediction performance comes close to unconstrained GAIL baselines (Sec. 5).

## 2 Setting and Problem Formulation

**Setting and definitions**

- We consider a dynamical system consisting of: *states* $s_t \in S$, at *time stages* $t \in 1{:}T := \{1, \ldots, T\}$;

- an *ego agent* that takes *action* $a_t \in A \subset \mathbb{R}^n, n \in \mathbb{N}$, according to its *ego policy* $\pi_t \in \Pi_t$ (for convenience we may drop subscript $t$); we allow $\pi_t$ to be either a conditional (Lebesgue) density, writing $a_t \sim \pi_t(a|s_t)$, or deterministic, writing $a_t = \pi_t(s_t)$;

- the ego agent can be either the *demonstrator*, denoted by $\pi^D$ (a priori unknown); or the *imitator*, denoted by $\pi^{I,\theta}$, with $\theta$ its parameters;

- *other agents*, also interpretable as *noise/perturbations* (a priori unknown), with their *(joint) action* $o_t$ according to *others' policy* $\sigma_t \in \Phi_t$ (density/deterministic, analogous to ego);

- the system's *transition function* (environment) $f$, s.t.,

$$s_{t+1} = f(s_t, a_t, o_t). \tag{1}$$

($f$ is formally assumed to be known, but uncertainty can simply be encoded into $o_t$.)

- *As training data*, we are given a set of *demonstrator trajectories* of the form $(s_1, a_1), (s_2, a_2), \ldots, (s_T, a_T)$ of $\pi^D$'s episodic rollouts in an instance of this dynamical system including other agents/noise.

- We consider a *(momentary) imitation cost function* $c(s, a)$ which formalizes the similarity between imitator and demonstrator trajectories (details follow in Sec. 4.2 on the GAIL-based $c$), and *(momentary) safety cost function* $d(s)$; states $s$ for which $d(s) \leq 0$ are referred to as *momentarily safe* or *collision-free*.

- Notation: $P(\cdot)$ denotes probability; and $p_x(\cdot)$ the probability's density of a variable $x$, if it exists (we may drop subscript $x$).

---

[1]Note that our approach works with both, demonstrators that are already fully safe, and those that may be unsafe.
[2]Regarding the task of realistic modeling/simulation: Adding safety constraints can, in principle, lead to unrealistic biases in case of actually unsafe demonstrations. But the other extreme is pure IL where, e.g., collisions rates can be unrealistically high (Bansal et al., 2018). So this is always a trade-off.

**Goal formulation** The goal is for the imitator to generate trajectories that minimize the expected imitation cost, while satisfying a safety cost constraint, formally:

$$\pi^I = \arg\min_{\pi} \underbrace{\mathbb{E}_{\pi}\left(\sum_{t=1}^{T} c(s_t, a_t)\right) - \psi(\pi)}_{=:\ v^{\pi,c,\psi}}, \qquad d(s_t) \leq 0, \text{ for all } t, \tag{2}$$

where $v^{\pi,c,\psi}$ is called *total imitation cost function* (we drop superscript $\psi$ if it is zero), $\psi(\pi)$ is a *policy regularizer*, and $\mathbb{E}_{\pi}(\cdot)$ denotes the expectation over roll-outs of $\pi$.[3] We purportedly stay fairly abstract here. We will later instantiate $c$, $\psi$ (Sec. 4.2) and $d$ (Sec. 5), and specify our safety-relevant modeling/reasoning about the other agents/perturbations (Sec. 3.1).

# 3 General Theoretical Tools for Policy and Training of Safe Generative Imitation Learning

We start by providing, in this section, several general-purpose theoretical foundations for safe generative IL, as tools to help the design of the safe imitator *policy* (Sec. 3.1 and 3.2), as well as to understand the advantages of end-to-end *training* (Sec. 3.3). We will later, in Sec. 4, build our method on these foundations.

## 3.1 Sample-based Inference of Safe Action Sets

For the design of safe imitator policies, it is helpful to know, at each stage, a *set of safe actions*. Roughly speaking, we consider an individual action as safe, if, after executing this action, at least *one safe* "best-case" future ego policy $\pi$ exists, i.e., keeping safety cost $d(s_t)$ below 0 for all $t$, under *worst-case* other agents/perturbations $\sigma$. So, formally, we define the *safe (action) set* in state $s$ at time $t$ as

$$\bar{A}_t^s := \{a \in A : \text{ it exists } \pi_{t+1:T}, \text{ s.t. for all } \sigma_{t:T} \text{ and for all } t < t' \leq T, \ d(s_{t'}) \leq 0 \text{ holds, given } s_t = s, a_t = a\} \tag{3}$$

(based on our general setting of Sec. 2). We refer to actions in $\bar{A}_t^s$ as *safe actions*. Such an adversarial/worst-case uncertainty reasoning resembles common formulations from reachability game analysis (Fridovich-Keil and Tomlin, 2020; Bansal et al., 2017). Furthermore, define the *total safety cost (to go)* as[4]

$$w_t(s, a) := \min_{\pi_{t+1:T}} \max_{\sigma_{t:T}} \max_{t' \in t+1:T} d(s_{t'}), \text{ for all } t. \tag{4}$$

Observe the following:

- The total safety cost $w_t$ allows us to quantitatively characterize the safe action set $\bar{A}_t^s$ as *sub-zero set*[5]:

$$\bar{A}_t^s = \{a : w_t(s, a) \leq 0\}. \tag{5}$$

- Under the worst-case assumption, for each $t, s$: an ego policy $\pi_t$ can be part of a candidate solution of our basic problem, i.e., satisfy safety constraint in Eq. (2), *only if $\pi_t$'s outputted actions are all in $\bar{A}_t^s$* (because otherwise there exists no feasible continuation $\pi_{t+1:T}$, by definition of $\bar{A}_t^s$). This fact will be crucial to define our safety layer (Sec. 3.2).

Now, regarding *inference* of the safe action set $\bar{A}_t^s$: while in certain limited scenarios, such as linear dynamics, it may be possible to efficiently calculate it exactly, this is not the case in general (Rungger and Tabuada, 2017). We take an approach to circumvent this by checking $w_t(s, a)$ for just a *finite sample* of $a$'s. And

---

[3]As usual, the arg min is well defined under appropriate compactness and continuity assumptions. The "=" may strictly be a "$\in$". We assume an initial (or intermediate) state as given.

[4]Note: In the scope of safety definitions, we generally let ego/other policies $\pi, \sigma$ range over compact sets $\Pi, \Phi$ of deterministic policies. Then the minima/maxima are well-defined, once we make appropriate continuity assumptions. The finite horizon can be extended to the infinite case using terminal safety terms Pek and Althoff (2020).

[5]Note that requiring momentary safety cost $d(s_{t'}) \leq 0$ *for each* future $t'$ in Eq. (3) translates to requiring that $d(s_{t'}) \leq 0$ for the *maximum* over future $t'$ in Eq. (4) (and not the sum over $t'$ as done in other formulations).

then concluding on the value of $w_t(s, \cdot)$ on these $a$'s "neighborhoods". This at least gives a so-called *inner approximation $\tilde{A}_t^s$ of the safe set $\bar{A}_t^s$*, meaning that, while $\tilde{A}_t^s$ and $\bar{A}_t^s$ may not coincide, we know for sure that $\tilde{A}_t^s \subset \bar{A}_t^s$, i.e, $\tilde{A}_t^s$ is safe. The following result gives us Lipschitz continuity and constant to draw such conclusions.[6] It builds on the fact that maximization/minimization preserves Lipschitz continuity (Appendix A.1.1). We assume all spaces are implicitly equipped with norms.

**Proposition 1** (Lipschitz constants for Lipschitz-based safe set)**.** *Assume the momentary safety cost $d$ is $\alpha$-Lipschitz continuous. Assume that for all (deterministic) ego/other policies $\pi_t \in \Pi_t, \sigma_t \in \Phi_t, t \in 1{:}T$, the dynamics $s \mapsto f(s, \pi_t(s), \sigma_t(s))$ as well as $a \mapsto f(s, a, \sigma_t(s))$ for fixed $s$ are $\beta$-Lipschitz. Then $a \mapsto w_t(s, a)$ is $\alpha \max\{1, \beta^T\}$-Lipschitz.*

Let us additionally give a second approach for sample-based inner approximation of the safe set.[7] The idea is that sometimes, safety of a finite set of *corners/extremalities* that span a set already implies safety of the *full* set:

**Proposition 2** (Extremality-based safe set)**.** *Assume the dynamics $f$ is linear, and the ego/other policy classes $\Pi_{1:T}, \Phi_{1:T}$ consist of open-loop policies, i.e., action trajectories, and that safety cost $d$ is convex. Then $a \mapsto w_t(s_t, a)$ is convex. In particular, for any convex polytope $B \subset A$, $w_t(s, \cdot)$ takes its maximum at one of the (finitely many) corners of $B$.*

Later we will give one Lipschitz-based method version (FAGIL-L) building on Prop. 1 (not using Prop. 2), and one convexity-based version (FAGIL-E) building on Prop. 2. For several broader remarks on the elements of this section, which are not necessary to understand this main part though, see also Appendix D.1.

## 3.2 Piecewise Diffeomorphisms for Flexible Safety Layers with Differentiable Closed-Form Density

An important tool to *enforce safety* of a policy's actions are safety layers. Assume we are given $\tilde{A} \subseteq \bar{A}$ as an (inner approximation of the) safe action set (e.g., from Sec. 3.1). A *safety layer* is a neural net layer that maps $A \to \tilde{A}$, i.e., it constrains a *"pre-safe"*, i.e., potentially unsafe, action $\hat{a} \in A$ into $\tilde{A}$. In contrast to previous safety layers (Donti et al., 2021a; Dalal et al., 2018) for deterministic policies, for our generative/probabilistic approach, we want a safety layer that gives us an overall policy's *closed-form differentiable density* (when plugging it on top of a closed-form-density policy like a Gaussian or normalizing flow), for end-to-end training etc. In this section we introduce a new function class and density formula that helps to construct such safety layers.

Remember the classic *change-of-variables formula* (Rudin et al., 1964): If $y = e(x)$ for some diffeomorphism[8] $e$, and $x$ has density $p_x(x)$, then the implied density of $y$ is $|\det(J_{e^{-1}}(y))|p_x(e^{-1}(y))$.[9] The construction of safety layers with exact density/gradient *only* based on change of variables can be difficult or even impossible. One intuitive reason for this is that diffeomorphisms require *injectivity*, and therefore one cannot simply map unsafe actions to safe ones and simultaneously leave safe actions where they are (as done, e.g., by certain projection-base safety layers (Donti et al., 2021a)).

These limitations motivate our simple yet flexible approach of using the following type of function as safety layers, which relaxes the rigid injectivity requirement that pure diffeomorphisms suffer from:

**Definition 1** (Piecewise diffeomorphism safety layers)**.** *We call a function $g : A \to \tilde{A}$ a piecewise diffeomorphism if there exists a countable partition $(A_k)_k$ of $A$, and diffeomorphisms (on the interiors) $g_k : A_k \to \tilde{A}_k \subset \tilde{A}$, such that $g|_{A_k} = g_k$.*

Now we can combine the diffeomorphisms' change-of-variables formulas with additivity of measures:

---

[6]Once we know that $w_t(s, \cdot)$ is $\gamma$-Lipschitz, and that $w_t(s, a) < 0$, then we also know that $w_t(s, \cdot)$ is negative on a ball of radius $\frac{|w_t(s,a)|}{\gamma}$ around $a$.

[7] Note that Gillula et al. (2014) also inner-approximate safe sets from finite samples of the current action space, by using the *convex hull* argument. But, among other differences, they do not allow for other agents, as we do in Prop. 2.

[8]A *diffeomorphism $e$* is a continuously differentiable bijection whose inverse is also cont. differentiable.

[9]$J_e$ denotes the Jacobian matrix of $e$.

**Proposition 3** (Closed-form density for piecewise diffeomorphism). *If $g$ is such a piecewise diffeomorphism, $\bar{a} = g(\hat{a})$ and $\hat{a}$'s density is $p_{\hat{a}}(\hat{a})$, then $\bar{a}$'s density is*

$$p_{\bar{a}}(\bar{a}) = \sum_{k:\bar{a}\in g_k(A_k)} |\det(J_{g_k^{-1}}(\bar{a}))| p_{\hat{a}}(g_k^{-1}(\bar{a})). \tag{6}$$

Specific instances of such piecewise diffeomorphism safety layers will be given in Sec. 4. Note that one limitation of piecewise diffeomorphism layers lies in them being discontinuous, which may complicate training. Further details related to diffeomorphic layers, which are not necessary to understand this main part though, are in Appendix D.2.

### 3.3 Analysis of End-to-End Training with Safety Layers vs. Test-Time-Only Safety Layers

After providing tools for the design of safe *policies*, now we focus on the following general question regarding *training* of such policies: *What is the difference, in terms of imitation performance, between using a safety layer only during test time, compared to using it during training and test time?* (Note: this question/section is a *justification*, but is not strictly necessary for understanding the *definition* of our method in Sec. 4 below. So this section may be skipped if mainly interested in the specific method and experiments.)

The question is important for the following reason: While we in this paper advocate end-to-end training including safety layer, in principle one could also use the safety layer *only* during test/production time, by composing it with an "unsafely trained" policy. This is nonetheless safe, but much easier computationally in training. The latter is an approach that other safe IL work takes (Chen et al., 2019), but it means that the *test-time policy differs from the trained one.*

While this question is, of course, difficult to answer in its full generality, let us here provide a theoretical analysis that improves the understanding at least under certain fairly general conditions. Our analysis is inspired by the study of the *compounding errors* incurred by behavior cloning (BC) (Ross and Bagnell, 2010; Syed and Schapire, 2010; Xu et al., 2020) For this section, let us make the following assumptions and definitions (some familiar from the mentioned BC/IL work):[10]

• State set $S$ and action set $A$ are finite. As usual (Ho and Ermon, 2016), the *time-averaged state-action distribution* of $\pi$ is defined as $\rho(s,a) := \left(\frac{1}{T}\sum_{t=1}^{T} p_{s_t}(s)\right)\pi(a|s)$, for all $s,a$; and $\rho(s)$ denotes the marginal.

• We assume some safety layer (i.e., mapping from action set into safe action set) to be given and fixed. As before, $\pi^D$ is the demonstrator, $\pi^I$ the *imitator trained with train-and-test-time safety layer*. Additionally define $\pi^U$ as a classic, *unsafe/unconstrained trained imitator* policy[11], and $\pi^O$ as the *test-time-only-safety policy* obtained by concatenating $\pi^U$ with the safety layer at test time. Let $\rho^D, \rho^I, \rho^U, \rho^O$ denote the corresponding time-averaged state-action distributions.

• As is common (Ross and Bagnell, 2010; Xu et al., 2020), we measure performance deviations in terms of the (unknown) demonstrator cost function, which we denote by $c^*$ (i.e., in the fashion of inverse reinforcement learning (IRL), where the demonstrator is assumed to be an optimizer of Eq. (2) with $c^*$ as $c$); and assume $c^*$ only depends on $s$, and $\|c^*\|_\infty$ is its maximum value. Let $v^D := v^{\pi^D,c^*}$, $v^I := v^{\pi^I,c^*}$, $v^O := v^{\pi^O,c^*}$ be the corresponding total imitation cost functions (Sec. 2).

• We assume that (1) the demonstrator always acts safely, and (2) the safety layer is the non-identity only on actions that are never taken by the demonstrator (mapping them to safe ones). Importantly, note that *if we relax these assumptions, in particular, allow the demonstrator to be unsafe, the results in this section would get even stronger* (the difference of adding a safety layer would be even bigger).

Keep in mind that what we can expect to achieve by GAIL training is a population-level closeness of imitator to demonstrator state-action distribution of the form $D(\rho^I, \rho^D) \le \varepsilon$, for some $\varepsilon$ decreasing in the sample size (due to the usual bias/generalization error; (Xu et al., 2020)), and $D$, e.g., the Wasserstein distance (Sec. 4.2). So we make this an assumption in the following results. Here we use $D(\rho^I, \rho^D) = D_{\mathrm{TV}}(\rho^I, \rho^D) = $

---

[10]We believe that most of the simplifying assumptions we make can be relaxed, but the analysis will be much more involved.

[11]In our case this would essentially mean to take the pre-safe policy, e.g., Gaussian, with pre-safe $\hat{a}_t$, and train it, as $\pi^U$.

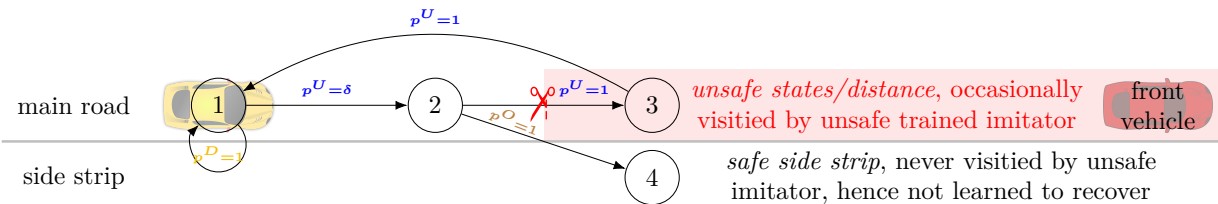

Figure 2: Staying with our running example of driver imitation, this figure illustrates the following: if we use safety layers only at test time and not already during training, the test-time imitator has *not learned to deal with the states that the safety layer may lead to* (i.e. to either plan to avoid, or recover from them), yielding *poor imitation performance*. In a nutshell: The demonstrator is always ($p^D = 1$) in state 1, meaning a constant velocity and distance to the front vehicle. At test/deployment time, the unsafely trained imitator $\pi^U$, due to learning error, with low probability $p^U = \delta$, deviates from state 1 and reaches first state 2 and then (due to, say, inertia) unsafe state 3, from which it has learned to always recover though. The test-time-only-safe imitator $\pi^O$, like the unsafely trained imitator it builds upon, with low probability $p^O = \delta$ reaches state 2. But from there, the *safety constraints (red scissors symbol) kick in* and the only remaining action is to go to the safe side strip 4 with $p^O = 1$, where there was *no data on how to recover (never visited/learned by unsafe $\pi^U$)*, getting trapped there forever. Details are in Appendix A.3.

$\frac{1}{2} \sum_{s \in S, a \in A} |\rho^I(s, a) - \rho^D(s, a)|$, the *total variation distance* (Klenke, 2013), which simplifies parts of the derivation, though we believe that versions for other $D$ such as Jensen-Shannon are possible.

Now, first, observe that for our "train *and* test safety layer" approach, we can have a *linear, in the horizon $T$*, imitation performance guarantee. This is because the *imitator $\pi^I$ is explicitly trained to deal with the safety layer* (i.e., to plan to avoid safe but poor states, or to at least recover from them; see Appendix A.3.1 for the derivation):

**Remark 1** (Linear error in $T$ of imitator with train-and-test-time safety layer)**.** *Assume $D_{TV}(\rho^I, \rho^D) \le \varepsilon$. Then we get*

$$|v^I - v^D| \le 2\varepsilon T \|c^*\|_\infty.$$

In contrast, using test-time-only safety, the test-time imitator $\pi^O$ *has not learned to plan with, and recover from, the states that the safety layer may lead to*. This leads, essentially, to a tight *quadratic* error bound (when accounting for arbitrary, including "worst", environments; proof in Appendix A.3):

**Theorem 1** (Quadratic error in $T$ of imitator with test-time-only safety layer)**.** ***Lower bound*** *(an "existence" statement): We can construct an environment[12] with variable horizon $T$ and with a demonstrator, sketched in Fig. 2 and additional details in Appendix A.3.2, a universal constant $\iota$, and, for every $\varepsilon > 0$, an unconstrainedly trained imitator $\pi^U$ with $D_{TV}(\rho^D, \rho^U) \le \varepsilon$, such that for the induced test-time-only-safe imitator $\pi^O$ we have, for all $T \ge 2$[13],*

$$|v^O - v^D| \ge \iota \min\{\varepsilon T^2, T\} \|c^*\|_\infty. \tag{7}$$

***Upper bound*** *(a "for all" statement): Assume $D_{TV}(\rho^D, \rho^U) \le \varepsilon$ and assume $\rho^U(s)$ has support wherever $\rho^D(s)$ has. Then*

$$|v^O - v^D| \le \frac{4\varepsilon}{\nu} T^2 \|c^*\|_\infty, \tag{8}$$

*where $\nu$ is the minimum mass of $\rho^D(s)$ within the support of $\rho^D(s)$.*

---

[12]In this discrete settings, the environment amounts to a Markov decision process (MDP) (Sutton and Barto, 2018).

[13]The quadratic bound in Eq. (7) is substantial once we look at small $\varepsilon$, because then the quadratic bound holds even for large $T$, while the train-and-test-time safety layer in Rem. 1 keeps scaling linearly in $T$. See also Rem. 4 in Appendix A.3.2.

# 4 Method Fail-Safe Adversarial Generative Imitation Learner

Building on the above theoretical tools and arguments for end-to-end training, we now describe our modular methodology FAGIL for safe generative IL. Before going into detail, note that, for our general continuous setting, tractably inferring safe sets and tractably enforcing safety constraints are both commonly known to be challenging problems, and usually require non-trivial modeling trade-offs (Rungger and Tabuada, 2017; Gillula et al., 2014; Achiam et al., 2017; Donti et al., 2021b). Therefore, here we choose to provide (1) one tractable *specific method instance* for the *low-dimensional* case (where we can efficiently partition the action space via a grid), which we also use in the experiments, alongside (2) an *abstract template* of a *general approach*. We discuss policy, safety guarantees (Sec. 4.1), imitation loss and training (Sec. 4.2).

## 4.1 Fail-Safe Imitator Policy with Invariant Safety Guarantee

First, our method consists of the *fail-safe (generative) imitator* $\pi^{I,\theta}(a|s)$. Its structure is depicted on the l.h.s. of Fig. 1, and contains the following modules: at each time $t$ of a rollout, with state $s_t$ as input,

**(a)** the **pre-safe generative policy module** outputs a pre-safe (i.e., potentially unsafe) sample action $\hat{a}_t \in A$ as well as explicit probability density $p_{\hat{a}|s}(\hat{a}|s_t)$ with parameter $\theta$ and its gradient. **Specific instantiations:** This can be, e.g., a usual Gauss policy or conditional normalizing flow.[14]

**(b)** The **safe set inference module** outputs an inner-approximation $\tilde{A}_t^{s_t}$ of the safe action set, building on our results form Sec. 3.1. The rough idea is as follows (recall that an action $a$ is safe if its total safety cost $w_t(s_t, a) \leq 0$): We check[15] $w_t(s_t, a)$ for a *finite sample* of $a$'s, and then conclude on the value of $w_t(s_t, \cdot)$ on these $a$'s *"neighborhoods"*, via Prop. 1 or Prop. 2.[16]

**Specific instantiations:** We partition the action set $A$ into boxes (hyper-rectangles) based on a regular grid (assuming $A$ is a box). Then we go over each box $A_k$ and determine its safety by evaluating $w_t(s_t, a)$, with either $a$ the center of $A_k$, and then using the Lipschitz continuity argument (Prop. 1) to check if $w_t(s_t, \cdot) \leq 0$ on the full $A_k$ (*FAGIL-L*), or $a$ ranging over all corners of box $A_k$, yielding safety of this box iff $w_t(s_t, \cdot) \leq 0$ on all of them (*FAGIL-E*; Prop. 2). Then, as inner-approximated safe set $\tilde{A}_t^{s_t}$, return the union of all safe boxes $A_k$.

**(c)** Then, pre-safe action $\hat{a}_t$ plus the safe set inner approximation $\tilde{A}_t^{s_t}$ are fed into the **safety layer**. The idea is to use a piecewise diffeomorphism $g$ from Sec. 3.2 for this layer, because the "countable non-injectivity" gives us quite some flexibility for designing this layer, while nonetheless we get a closed-form density by summing over the change-of-variable formulas for all diffeomorphisms that map to the respective action (Prop. 3). As output, we get a sample of the safe action $\bar{a}_t$, as well as density $\pi^{I,\theta}(\bar{a}_t|s_t)$ (by plugging the pre-safe policy's density $p_{\hat{a}|s}(\hat{a}|s_t)$ into Prop. 3) and gradient w.r.t. $\theta$.

**Specific instantiations:** We use the partition of $A$ into boxes $(A_k)_k$ from above again. The piecewise diffemorphism $g$ is defined as follows, which we believe introduces a helpful inductive bias: (a) On all safe boxes $k$, i.e., $A_k \subset \bar{A}$, $g_k$ is the identity, i.e., we just keep the pre-safe proposal $\hat{a}$. (b) On all unsafe boxes $k$, $g_k$ is the translation and scaling to the most nearby safe box in Euclidean distance. Further safety layer instances and illustrations are in Appendix D.3.

**Adding a safe fallback memory**  To cope with the fact that it may happen that at some stage an *individual* safe action exists, but no *fully* safe part $A_k$ exists (since the $A_k$ are usually sets of more than one action and not all have to be safe even though some are safe), we propose the following, inspired by (Pek and Althoff, 2020): the fail-safe imitator has a *memory* which, at each stage $t$, contains one purportedly *fail-safe fallback future ego policy* $\pi_{t+1:T}^f$. Then, if no fully safe part $A_k$ (including a new fail-safe fallback) is found anymore at $t+1$, execute the fallback $\pi_{t+1}^f$, and keep $\pi_{t+2:T}^f$ as new fail-safe fallback (see also Appendices B and D.3.5 on potential issues with this).

---

[14]If $A$ is box-shaped, then we add a squashing layer to get actions into $A$ using component-wise sigmoids.

[15]Clearly, already the "checker" task of evaluating a single action's safety, Eq. (4), can be non-trivial. But often, roll-outs, analytic steps, convex optimizations or further inner approximations can be used, or combinations of them, see also Sec. 5.

[16]Due to modularity, other safety frameworks like Responsibility-Sensitive Safety (RSS) Shalev-Shwartz et al. (2017) can be plugged into our approach as well. The fundamental safety trade-off is between being safe (conservativity) versus allowing for enough freedom to be able to "move at all" (actionability).

**Remark 2** (Invariant safety guarantee)**.** *Based on the above, if we know at least one safe action at stage* 1*, then we invariably over time have at least one safe action plus subsequent fail-safe fallback policy. So we are* guaranteed *that the ego will be safe throughout the horizon* 1:$T$.

### 4.2 Imitation Cost and Training Based on Generative Adversarial Imitation Learning

In principle, various frameworks can be applied to specify imitation cost (which we left abstract in Sec. 2) and training procedure for our fail-safe generative imitator. Here, we use a version of GAIL (Ho and Ermon, 2016; Kostrikov et al., 2018) for these things. In this section, we give a rough idea of how this works, while some more detailed background and comments for the interested reader are in Appendix B. In GAIL, roughly speaking, the imitation cost $c(s,a)$ is given via a GAN's *discriminator* that tries to distinguish between imitator's and demonstrator's average state-action distribution, and thereby measures their dissimilarity. As policy regularizer $\psi(\pi)$ in Eq. (2), we take $\pi$'s differential entropy (Cover and Thomas, 2006).

**Training**   As illustrated on the r.h.s. of Fig. 1, essentially, training consists of steps alternating between: (a) doing rollouts of our fail-safe imitator $\pi^{I,\theta}(a|s)$ under current parameter $\theta$ to sample imitator state-action trajectories (the *generator*); (b) obtaining new discriminator parameters and thus $c$, by optimizing a discriminator loss, based on trajectory samples of both, imitator $\pi^I$ and demonstrator $\pi^D$; (c) obtaining new fail-safe imitator policy parameter $\theta'$ by optimizing an estimated Eq. (2) from the trajectory sample and $c$. Note that in (c), the fail-safe imitator's closed-form density formula (Prop. 3) and its gradient are used (the precise use depends on the specific choice of policy gradient (Kostrikov et al., 2018; Sutton and Barto, 2018)).

## 5 Experiments

The goal of the following experiments is to demonstrate tractability of our method, and empirically compare safety as well as imitation/prediction performance to baselines. We pick driver imitation from real-world data as task because it is relevant for both, *control* (of vehicles), as well as *modeling* (e.g., of real "other" traffic participants, for validation of self-driving "ego" algorithms Suo et al. (2021); Igl et al. (2022)). Furthermore, its characteristics are quite representative of various challenging safe control and robust modeling tasks, in particular including fixed (road boundary) as well as many moving obstacles (other drivers), and multi-modal uncertainties in driver behavior.

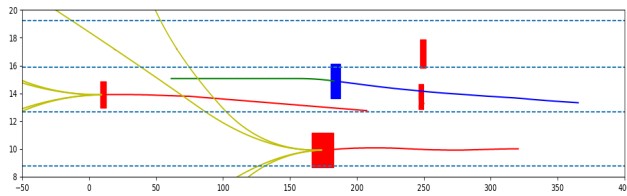

Figure 3: Driving from right to left, our fail-safe imitator (blue) with past trajectory (blue) and future fail-safe fallback emergency brake (green); other vehicles (red), with worst-case reachable boundaries indicated in yellow (rear vehicles are ignored as described).

**Data set and preprocessing**   We use the open *highD data set* (Krajewski et al., 2018), which consists of 2-D car trajectories (each ~20s) recorded by drones flying over highway sections (we select a straight section with ~ 1500 trajectories). It is increasingly used for benchmarking (Rudenko et al., 2019; Zhang et al., 2020). We filter out other vehicles in a roll-out once they are behind the ego vehicle.[17] Additionally, we filter out initially unsafe states (using the same filter for all methods).

**Setting and simulation**   We do an open-loop simulation where we only replace the ego vehicle by our method/the baselines, while keeping the others' trajectories from the original data. The action is 2-D (longitudinal and lateral acceleration) and as the simulation environment's dynamics $f$ we take a component-wise double integrator. As collision/safety cost $d(s)$, we take minus the minimum $l_\infty$ distance (which is

---

[17]This is substantial filtering, but is, in one way or another, commonly done (Pek and Althoff, 2020; Shalev-Shwartz et al., 2017). It is based on the idea that the ego is not blamable for other cars crashing into it from behind (and otherwise safety may become too conservative; most initial states could already be unsafe if we consider adversarial rear vehicles that deliberately crash from behind).

Table 1: Outcome on real-world driver data, in terms of imitation and safety (i.e., collision) performance.

| Pre-safe | Method Overall | Imitation performance | | Safety performance |
|---|---|---|---|---|
| | | ADE | FDE | Probability of crash/off-road |
| Gauss | FAGIL-E (ours) | 0.59 | 1.70 | 0.00 |
| | FAGIL-L (ours) | 0.60 | 1.77 | 0.00 |
| | GAIL Ho and Ermon (2016) | 0.47 | 1.32 | 0.13 |
| | RAIL Bhattacharyya et al. (2020) | 0.48 | 1.35 | 0.22 |
| | TTOS (Sec. 3.3) | 0.60 | 1.78 | 0.00 |
| Flow | FAGIL-E (ours) | 0.58 | 1.69 | 0.00 |
| | FAGIL-L (ours) | 0.57 | 1.68 | 0.00 |
| | GAIL Ho and Ermon (2016) | 0.44 | 1.22 | 0.11 |
| | RAIL Bhattacharyya et al. (2020) | 0.53 | 1.50 | 0.11 |
| | TTOS (Sec. 3.3) | 0.59 | 1.72 | 0.00 |

particularly easy to compute) of ego to other vehicles (axis-aligned bounding boxes) and road boundary, which is 2-Lipschitz (from norm $\| \cdot \|_\infty$ to $| \cdot |$).

**Policy fail-safe imitator** We use our FAGIL imitator policy from Sec. 4.1. As learnable *state feature*, the state is first rendered into an abstract birds-eye-view image, depicting other agents and road boundaries, which is then fed into a convolutional net (CNN) for dimensionality reduction, similar as described in (Chai et al., 2019). *Regarding safe set inference:* In this setting, where the two action/state dimensions and their dynamics are separable, the others' reachable rectangles can simply be computed exactly based on the others' maximum longitudinal/lateral acceleration/velocity (this can be extended to approximate separability (Pek and Althoff, 2020)). As ego's fallback maneuver candidates, we use non-linear shortest-time controllers to roll out emergency brake and evasive maneuver trajectories (Pek and Althoff, 2020). Then the total safety cost $w$ is calculated by taking the maximum momentary safety cost $d$ between ego's maneuvers and others' reachable rectangles over time. Note that the safety cost $d$ and dynamics $f$ with these fallbacks are each Lipschitz continuous as required by Prop. 1, rigorously implying the *safety guarantee* for FAGIL-L. (FAGIL-E does only roughly but not strictly satisfy the conditions of Prop. 2, but we conjecture it can be extended appropriately.) *As safety layer*, we use the distance-based instance from Sec. 4.1. **Training:** Training happens according to Sec. 4.2, using GAIL with soft actor-critic (SAC) (Kostrikov et al., 2018; Haarnoja et al., 2018) as policy training (part (c) in Sec. 4.2), based on our closed-form density from Sec. 3.2. Further details on setup, methods and outcome are in Appendix C.

**Baselines, pre-safe policies, ablations, metrics and evaluation** Baselines are: classic *GAIL* Ho and Ermon (2016), *reward-augmented GAIL (RAIL)* (closest comparison from our method class of safe generative IL, discussed in Sec. 1) (Bhattacharyya et al., 2020; 2019), and GAIL with *test-time-only safety layer (TTOS)* (Sec. 3.3). Note that TTOS can be seen as an *ablation study* of our method where we drop the safety layer during train-time, and GAIL an ablation where we drop it during train and test time. For all methods including ours, as *(pre-safe) policy* we evaluate both, *Gaussian policy* and *conditional normalizing flow*. We evaluate, on the test set, overall probability (frequency) of collisions over complete roll-outs (including crashes with other vehicles and going off-road) as *safety performance* measure; and *average displacement error (ADE*; i.e., time-averaged distance between trajectories) and *final displacement error (FDE)* between demonstrator ego and imitator ego over trajectories of 4s length as *imitation/prediction performance* measure (Suo et al., 2021).

**Outcome and discussion** The outcome is in Table 1 (rounded to two decimal points). It empirically validates our theoretical *safety* claim by reaching 0% collisions of our FAGIL methods (with fixed and moving

obstacles; note that additionally, TTOS validates safety of our safety inference/layer). At the same time FAGIL comes close in *imitation* performance to the baselines RAIL/GAIL, which, even if using reward augmentation to penalize collisions (RAIL), have significant collision rates. (The rather high collision rate of RAIL may also be due to the comparably small data set we use.) FAGIL's imitation performance is slightly better than TTOS, which can be seen as a hint towards an average-case confirmation, for this setting, of our theoretical worst-case analysis (Thm. 1), but the gap is rather narrow. Fig. 3 gives one rollout sample, including imitator ego's fail-safe and others' reachable futures considered by our safe set inference.

## 6  Conclusion

In this paper, we considered the problem of safe and robust generative imitation learning, which is of relevance for robust realistic simulations of agents, as well as when bringing learned agents into many real-world applications from robotics to highly automated driving. We filled in several gaps that existed so far towards solving this problem, in particular in terms of sample-based inference of guaranteed adversarially safe action sets, safety layers with closed-form density/gradient, and the theoretical understanding of end-to-end generative training with safety layers. Combining these results, we described a general abstract method for the problem, as well as one specific tractable instantiation for a low-dimensional setting. The latter we experimentally evaluated on the task of driver imitation from real-world data, which includes fixed and moving, uncertain obstacles (the other drivers).

Limitations of our work in particular come from the challenges that safety goals often bring with them, in terms of tractability and cautiousness: First, while our theoretical results hold more generally, the specific grid-based method instantiation we give is tractable only in the low-dimensional case, and in the experimental setting we harnessed tractability via separability of longitudinal and lateral dynamics of the other agents. Second, worst-case safety can lead to overly cautious actions while human "demonstrators" often achieve surprisingly good trade-offs in this regard; incorporating more restrictions about other agents, such as the responsibility-sensitive safety (RSS) framework, would still be covered by our general theory but help to be less conservative. We also inherit known drawbacks of generative adversarial imitation learning-based approaches, in particular instabilities in training, and this can further increase by the additional complexity induced by the safety layer. On the experimental evaluation side, we focused on a first study on one challenging but restricted setting, leaving an extensive purely empirical study to future work.

In this sense, our work constitutes one step on the path of combining *flexibility and scalability* (i.e., general capacity for learning with little hand-crafting) of neural net-based probabilistic policies with theoretic worst-case *safety guarantees*, while maintaining *end-to-end* generative trainability.

### Broader Impact Statement

When controlling agents in physical environments that also involve humans, such as robots that interact with humans, where even small and/or rare mistakes of methods can injur people, "pure" deep imitation learning approaches can be problematic. This is because often they are "black boxes" for which we do not understand precisely enough how they behave "outside the training distribution". Our approach can ideally have a positive impact on making imitation learning-based methods more safe and robust for such domains (or making them more applicable for such domains in the first place). One limitation of our work (beyond the technical limitations discussed above) is that in the end, society a has to debate and decide on the difficult trade-offs in terms of cautiousness versus actionability that safe control often involves (clearly safety is of highest value, but the question is how much residual risk society wants to take, given there rarely exists perfect safety). Science can just acknowledge the problem and provide understanding and tools. And overall, automation via machine learning approaches, including our approach, can have positive and negative impacts on jobs and the economoy overall, which society needs to consider and make decisions about.

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

# Appendix

# A    Proofs with remarks

This section contains all proofs for the mathematical results of the main text, as well as some additional remarks.

Note: In the main text, we did neither explicitly introduce the probaility space and random variables, nor explicitly distinguish values/events of such variables from the variables themselves, since this was not crucial and would have hindered the readability. However, in some of the following proofs, whenever it is necessary to have full rigor, we will make these things explicit.

## A.1    Proofs for Sec. 3.1

### A.1.1    Proof of Prop. 1

We will need the following known result (a fact that lies somewhere between *envelope* and *maximum theorem*):

**Fact 1.** *Let $h$ be a function from $X \times Y$ to $\mathbb{R}$, and assume $X$ and $\mathbb{R}$ to be equipped with some given norms (to which Lipschitz continuity refers in the following). Assume $h(x, y)$ is Lipschitz in $x$ with constant $L$ uniformly in $y$ and assume that maxima exist. Then the function $x \mapsto \max_y h(x, y)$ is also $L$-Lipschitz.*

For the sake of completeness, we provide a proof of Fact 1. Note: We show the statement for the case that a supremum is always taken, i.e., is a maximum. For the supremum version we refer the reader to related work.[18]

*Proof of Fact 1.* Let $\|\cdot\|$ and $|\cdot|$ denote the given norms on $X, \mathbb{R}$, respectively. Let $e(x) := \arg\max_y h(x, y)$ (or pick one argmax if there are several).

Consider an arbitrary but fixed pair $x, x'$ and let, w.l.o.g., $\max_y h(x, y) \geq \max_y h(x', y)$. Then:

$$| \max_y h(x, y) - \max_y h(x', y)| \tag{9}$$

$$= \max_y h(x, y) - \max_y h(x', y) \tag{10}$$

$$= h(x, e(x)) - h(x', e(x')) \tag{11}$$

$$\leq h(x, e(x)) - h(x', e(x)) \qquad (\text{since } h(x', e(x')) \geq h(x', e(x))) \tag{12}$$

$$\leq L\|x - x'\|. \tag{13}$$

$\square$

Now we can use this fact for the proof of Prop. 1:

*Proof for Prop. 1.* The idea is to first propagate the Lipschitz continuity through the dynamics, and then iteratively apply Fact 1 to the min/max operations.

Here, for any Lipschitz continuous function $e$, let $L_e$ denote its constant. Furthermore, regarding policies, we may write $\pi, \sigma$ as shorthand for $\pi_t, \sigma_t, t \in 1:T$, respectively.

First observe the following, given an arbitrary but fixed $t$ and $\pi, \sigma$:

Let

$$g(s, \pi, \sigma) := f(s, \pi(s), \sigma(s)) \tag{14}$$

---

[18]For a proof, see e.g., `https://math.stackexchange.com/q/2532116/1060605`.

and $h(s, \pi, \sigma, k)$ be defined as the $k$-times concatenation of $g(\cdot, \pi, \sigma)$, i.e., a roll out; and then define function $i$ by also including the initial action, i.e.,

$$i(s, a, \pi, \sigma, k) := h(f(s, a, \sigma(s)), \pi, \sigma, k). \tag{15}$$

Then, by our assumptions, for $\pi, \sigma, k$ arbitrary but fixed,

$$s \mapsto d(h(s, \pi, \sigma, k)) \tag{16}$$

is Lipschitz continuous in $s$ with constant (uniformly in $\pi, \sigma$)

$$L_d L_{f(\cdot, \pi(\cdot), \sigma(\cdot))}^k = \alpha \beta^k, \tag{17}$$

since it is the composition of $d$, which is $\alpha$-Lipschitz, with the $k$-times concatenation of $g$, which is $\beta$-Lipschitz by assumption. This together with our assumption implies that, for any arbitrary but fixed $s$, also

$$a \mapsto d(i(s, a, \pi, \sigma, k))(= w_t(s, a)) \tag{18}$$

is Lipschitz uniformly in $\pi, \sigma$, with constant (for any $k$)

$$\alpha \beta^k \beta = \alpha \beta^{k+1} \le \alpha \max\{1, \beta^T\}. \tag{19}$$

Now let state $s$ be arbitrary but fixed. Also let $\pi$ be arbitrary but fixed. Then, based on the above together with Fact 1,

$$a \mapsto \max_{\sigma_{t:T}} \max_{t \in t+1:T} d(i(s, a, \pi, \sigma, t))(=: e(a, \pi)) \tag{20}$$

is Lipschitz with constant $\alpha \max\{1, \beta^T\}$. Again applying Fact 1, we see that also

$$a \mapsto \min_{\pi_{t+1:T}} \max_{\sigma_{t:T}} \max_{t \in t+1:T} d(i(s, a, \pi, \sigma, t))(= \min_\pi e(a, \pi)) \tag{21}$$

is Lipschitz with constant $\alpha \max\{1, \beta^T\}$.

$\square$

**Remark 3.** *Alternative versions of this result could be based, e.g., on the "Envelope Theorem for Nash Equilibria" (Caputo, 1996).*

### A.1.2 Proof of Prop. 2

*Proof for Prop. 2.* Let the mapping $i$ be defined as in Eq. (15). Fix $s$.

Observe that, based on our assumption that $d$ is convex and the dynamics $f$ linear, for any $\sigma, t$ fixed,

$$(a, \pi) \mapsto d(i(s, a, \pi, \sigma, t)) \tag{22}$$

is convex. But then, also the maximum over $\sigma, t$, i.e.,

$$(a, \pi) \mapsto \max_{\sigma_{t:T}} \max_{t \in t+1:T} d(i(s, a, \pi, \sigma, t)) \tag{23}$$

is convex, because the element-wise maximum over a family of convex functions – in our case the family indexed by $\sigma, t$ – is again convex.

One step further, this means (since convexity is preserved by "minimizing out" one variable over a convex domain) that also taking the minimum over $\pi$ preserves convexity, i.e.,

$$w_t(s, a) = \min_{\pi_{t+1:T}} \max_{\sigma_{t:T}} \max_{t \in t+1:T} d(i(s, a, \pi, \sigma, t)) \tag{24}$$

is convex as a function of $a$.

But a convex function over a (compact) convex set that is spanned by a set of corners, takes its maximum at (at least one) of the corners.

$\square$

## A.2 Proofs for Sec. 3.2

### A.2.1 Proof of Prop. 3

*Proof for Prop. 3.* Intuitively it is clear that the results follows from combining $\sigma$-additivity of measures with the change-of-variables formula. Nonetheless we give a rigorous derivation here.

Keep in mind the following explicit specifications that were left implicit in the main part: In the main part we just stated to assume Lebesgue densities on $A \subset \mathbb{R}^n$, implicitly we assume $\mathbb{R}^n$ with the usual Lebesgue $\sigma$-algebra as underlying measurable space, and that all parts $A_k$ are measurable. Furthermore, to be precise, we assume differentiability of the $g_k$ on the *interior* of their respective domain.

Recall that $(A_k)_k$ is a countable partition of $A$, and we assumed diffeomorphisms $g_k : A_k \to \bar{A}_k \subset \tilde{A}$ (diffeomorphic on the interior of $A_k$) and that $g|_{A_k} = g_k$.

Note that based on our assumptions, either $g$ is already measurable, or, otherwise, we can turn it into a measurable function by just modifying it on a Lebesgue null set (the boundaries of the $A_k$), not affecting the below argument.

Let $P_A$ denote the original measure on outcome space $A$ (restriction from $\mathbb{R}^n$), and $P_{\tilde{A}}$ its push-forward measure on $\tilde{A}$ induced by $g$.

Then, up to null sets, for any measurable set $M \subset \tilde{A}$,

$$P_{\tilde{A}}(M) \tag{25}$$

$$= P_A(g^{-1}(M)) \tag{26}$$

$$= P_A(g^{-1}(M) \cap \bigcup_k A_k) \tag{27}$$

$$= P_A(\bigcup_k g^{-1}(M) \cap A_k) \tag{28}$$

$$= P_A(\bigcup_k g_k^{-1}(M)) \tag{29}$$

$$= \sum_k P_A(g_k^{-1}(M)) \tag{30}$$

$$= \sum_k \int_M |\det(J_{g_k^{-1}}(\bar{a}))| p_{\hat{a}}(g_k^{-1}(\bar{a}))[\bar{a} \in g_k(A_k)] d\bar{a} \tag{31}$$

$$= \int_M \sum_{k:\bar{a} \in g_k(A_k)} |\det(J_{g_k^{-1}}(\bar{a}))| p_{\hat{a}}(g_k^{-1}(\bar{a})) d\bar{a} \tag{32}$$

where Eq. (31) is the classic change-of-variables (also called transformation formula/theorem) (Klenke, 2013) applied to the push-forward measure of $P_A$ under $g_k$. Therefore, the[19] density of $P_{\tilde{A}}$, i.e., $p_{\bar{a}}(\bar{a})$, is given by $\sum_{k:\bar{a} \in g_k(A_k)} |\det(J_{g_k^{-1}}(\hat{a}))| p_{\hat{a}}(g_k^{-1}(\bar{a}))$.

$\square$

## A.3 Proofs and precise counterexample figure/MDP/agents for Sec. 3.3

In this section we give all elaborations and proofs for the end-to-end versus test-time-only safety analysis of Sec. 3.3.

For this section, keep in mind the following notation and remarks:

- In the discrete setting of this section, the imitation cost function $c$ etc. are vectors in the Euklidean space here; and, for instance, $c \cdot P_{S_t}^D$ or just $cP_{S_t}^D$ stands for the *inner product*, the expectation of the cost variable $c$.

---

[19]Rigorously speaking, it is *a* density.

- If not stated otherwise, $\|\cdot\|$ refers to the $l_1$-norm $\|\cdot\|_1$.

- Keep in mind that for the total variation distance $D_{\text{TV}}$ we have the relation to the $l_1$ norm $D_{\text{TV}}(p,q) = \frac{1}{2}\|p-q\|_1$.

- Keep in mind that, as in the main text, letters $D, I, U, O$ stand for demonstrator, imitator (our fail-safe imitator with safety layer already during training time), unsafe/unconstrained trained imitator, and test-time-only-safety imitator, respectively; and $P^D, P^I, P^U, P^O$ etc. denote the probabilities under the respective policies, and same for $\rho^D$ etc.

### A.3.1 Derviation of Rem. 1

The following derivation works along the line of related established derivations, e.g., the one for infinite-horizon GAIL guarantees (Xu et al., 2020).

*Derviation of Rem. 1.* Keep in mind that for any discrete distributions $p_{s,a}(s,a), q_{s,a}(s,a)$, when marginal-izig out $a$, we get for $p_s, q_s$,

$$\|p_s - q_s\| \tag{33}$$

$$= \sum_s |p(s) - q(s)| \tag{34}$$

$$= \sum_s |\sum_a p(s,a) - q(s,a)| \tag{35}$$

$$\leq \sum_s \sum_a |p(s,a) - q(s,a)| \qquad \text{(triangle inequality)} \tag{36}$$

$$= \|p_{s,a} - q_{s,a}\|. \tag{37}$$

Now we have

$$|v^I - v^D| \tag{38}$$

$$= |\sum_t cP^I_{s_t} - \sum_t cP^D_{s_t}| \tag{39}$$

$$= |cT(\sum_t \frac{1}{T}P^I_{s_t} - \sum_t \frac{1}{T}P^D_{s_t})| \tag{40}$$

$$= T|c(\rho^{\text{I}}_s - \rho^{\text{D}}_s)| \qquad \text{(Definition } \rho) \tag{41}$$

$$\leq T\|c\|_\infty \|\rho^{\text{I}}_s - \rho^{\text{D}}_s\| \tag{42}$$

$$\leq T\|c\|_\infty \|\rho^{\text{I}}_{s,a} - \rho^{\text{D}}_{s,a}\| \qquad \text{(Eq. (37))} \tag{43}$$

$$= T\|c\|_\infty 2D_{\text{TV}}(\rho^{\text{I}}_{s,a}, \rho^{\text{D}}_{s,a}). \tag{44}$$

$$\square$$

### A.3.2 Proof of Thm. 1, including elaboration of example (figure)

**Elaboration of the example for the lower bound (Fig. 2)** Here, let us first give a precise version – Fig. 4 – of the example in Fig. 2, that is used to get the lower bound in Thm. 1. Instead of the exact Fig. 2, we give an "isomorphic" version with more detailed/realistic states and actions in terms of relative position/velocity/accleration/lane.

Specify the Markov decision process (MDP) and agents (policies) as follows, with imitation error $p^U = \delta$ as paramteter:

**MDP:**

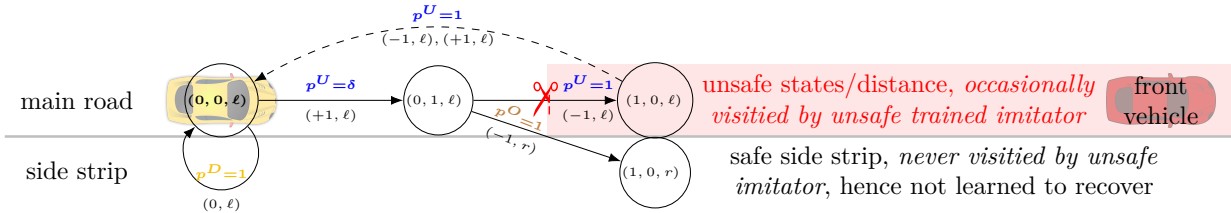

Figure 4: More detailed (but essentially isomorphic) version of the example MDP/agents from Fig. 2.

- *States:* The states (of the ego) are triplets ($\Delta x$, $\Delta v$, lane), where $\Delta x$ is deviation from demonstrator position, $\Delta v$ is deviation from demonstrator velocity, lane is lane ($\ell$ for left, or $r$ for right). (The demonstrator is in a safe state.) There are five possible states, all depicted in the figure, except for $(1, -1, \ell)$, to keep the figure simple.

- *Actions:* The possible actions are depicted as arrows, with the action (acc, lane) written below the arrow, where acc means the longitudinal acceleration, and lane the target lane.

- *Transition:* The longitudinal transition is the usual double integrator of adding acc to $\Delta v$, while, lateral-wise, action lane instantaneously sets the new lane (more realistic but complex examples are possible of course).

- *Costs:* The true demonstrator's cost $c^*$ is 0 on the demonstrator state $(0, 0, \ell)$, and 1 on every other state.

- *Horizon:* form $t = 1$ to $T$, where we consider $T$ a parameter (so it is strictly speaking a family of MDPs).

**Agents:**

- *Demonstrator:* Yellow car, always ($p^D = 1$) staying at the depicted state, referred to as $s^D$ ($= (0, 0, \ell)$), at a more than safe distance to front car (red).

- *Unconstrained trained imitator (blue):* due to a slight imitation error, it is sometimes (with probability $p^U = \delta$) accelerating, and if it accelerates, it necessarily reaches an unsafe state (after one more step, due to the "inertia" of the double integrator), but, as implied by the GAIL loss, always recovers (dashed arrows) back to true demonstrator state.

- *Safety-constrained test-time imitator (same, except for last action in brown):* as the unconstrained imitator, it occasionally ($p^O = \delta$) ends up at rightmost safe state, *and then the only remaining safe action is to change lane to the side strip, where there was no data on how to recover, so getting trapped there forever.*

**Initialization − agent-dependent:** While the overall agrument works as well for an agent-independent initialization, we do an agent-specific initialization, since it simplifies calculations substantially. At $t = 1$, for the demonstartor, we let $p(s_1)$ have full mass on the demonstrator state $s^D$. For the unconstrained imitator, as $p(s_1)$, we take the stationary distribution under the induced Markov process, which is $\frac{1}{1+3\delta}(1, \delta, \delta, \delta)$, where the first component means the demonstartor state $s^D$.

**Proof of Thm. 1** Keep in mind that we will occasionally use upper case letters like $S_t, A_t$, etc., to refer to the actual random variables of state and action, etc., distinguishing them from the values $s_t, a_t$ that those random variables can take.

*Proof for Thm. 1.* **Lower bound part:**

Keep in mind the counterexample construction above, in particular its deviation parameter $\delta$, to which the following arguments refer. Let $s^D, a^D$ denote the state and action that the demonstrator is constantly in.

*Construction of $\delta$; implication $D_{TV}(\rho^D, \rho^U) \leq \varepsilon$:*

By construction, we have $\rho^D(s^D, a^D) = 1$, while for the unconstrained imitator, $\rho^U(s^D, a^D) = \frac{1}{1+3\delta} \cdot (1 - \delta)$ (recall that $\rho^U(s)$ is the time-average over the state distribution, which we took as the stationary one for the unconstrained imitator, giving mass $\frac{1}{1+3\delta}$ on $s^D$, and $1 - \delta$ is the action probability).

So

$$|\rho^D(s^D, a^D) - \rho^U(s^D, a^D)| = |1 - \frac{1-\delta}{1+3\delta}| = \frac{4\delta}{1+3\delta}.$$

So, by symmetry, $D_{\mathrm{TV}}(\rho^D, \rho^U)$ is one half times twice this,

$$D_{\mathrm{TV}}(\rho^D, \rho^U) \leq \frac{4\delta}{1+3\delta}.$$

Additionally, it is easy to see that for $1 \geq \delta \geq 0$, $\frac{4\delta}{1+3\delta} \leq 4\delta$, and so

$$D_{\mathrm{TV}}(\rho^D, \rho^U) \leq 4\delta. \tag{45}$$

Therefore choose

$$\delta := \frac{1}{4}\varepsilon, \tag{46}$$

based on the $\varepsilon$ given by assumption. This implies $D_{\mathrm{TV}}(\rho^D, \rho^U) \leq \varepsilon$.

*Preparation of a bound statement we need later:*

As preparation which we repeatedly need later, observe that there is some constant $\kappa > 0$, such that for all $T \geq 2$, we have

$$\kappa \leq (1 + \frac{-1}{T})^T \leq e^{-1}, \tag{47}$$

where $e$ here is the Euler number (exponential) as usual, and for $\kappa$ we can take, e.g., $\frac{1}{4}$. To have a *first hint* why this holds, see that

$$(1 + \frac{-1}{T})^T \xrightarrow{T \to \infty} e^{-1}.$$

For the *full argument*, observe the following: First, observe that at $T = 2$, we have $(1 + \frac{-1}{T})^T = \frac{1}{4}$. Second, it is sufficient to show that the expression is monotonically growing, i.e., the derivative $\frac{\mathrm{d}}{\mathrm{d}T}(1 + \frac{-1}{T})^T$ is positive for $T \geq 2$. But this is equivalent to $\frac{\mathrm{d}}{\mathrm{d}T} \log((1 + \frac{-1}{T})^T) = \frac{1}{T-1} + \log(\frac{T-1}{T})$ being positive, since log is a monotonic function. But the latter follows from the fact that always $x \geq \log(1 + x)$ and then plugging in $\frac{1}{T-1}$ for $x$, yieldig $\frac{1}{T-1} \geq \log(1 + \frac{1}{T-1}) = \log(\frac{T}{T-1}) = -\log(\frac{T-1}{T})$.

Now, before going into the main proof parts, recall that we want to show

$$|v^O - v^D| \geq \iota \min\{\varepsilon T^2, T\} \|c^*\|_\infty \tag{48}$$

for $\iota$ some constant indepentent of $\varepsilon, T$.

We do so by considering the case where it is quadratic separately from where it is linear.[20]

*Proof for the lower bound, case $\delta T \leq 1$:*

Consider the case $\delta T \leq 1$.

---

[20]Note that our construction is different from behavior cloning error analysis (Ross and Bagnell, 2010; Syed and Schapire, 2010) especially in that we do not need one separate starting state.

By construction, at each fixed stage $t$, the probability that the test-time constrained imitator $\pi^O$ is not in the demonstrator state is the summed probability of not deviating from $s^D$ for $k$ times, $k \leq t$, and then deviating once, i.e.,

$$P^O(S_t \neq s^D) = \sum_{k=1}^{t} \delta(1-\delta)^{k-1}. \tag{49}$$

We want to bound the term $(1-\delta)^{k-1}$ from below, to show that the quadratic bound of Eq. (48) holds under the current case $\delta T \leq 1$.

We have, for any $k \leq T$,

$$(1-\delta)^k \geq (1 - \frac{1}{T})^k \geq (1 - \frac{1}{T})^T = (1 + \frac{-1}{T})^T \geq \kappa \tag{50}$$

where the first inequality holds because we assumed $\delta \leq \frac{1}{T}$ and so $1 - \delta \geq 1 - \frac{1}{T}$, and the second one because always $k \leq T$ and third is Eq. (47).

Therefore Eq. (49) can be bounded from below by

$$\sum_{k=1}^{t} \delta\kappa = t\delta\kappa. \tag{51}$$

Now we get, with the cost vector $\vec{c}$ (i.e., the safety cost function $c$, but making explicit it is a vector since we are in the discrete case) being 0 in the demonstrator state, and $\|\vec{c}\|_\infty$ elsewhere,

$$|v^O - v^D| \tag{52}$$

$$= |\sum_t (P^O_{s_t} - P^D_{s_t}) \cdot \vec{c}| \tag{53}$$

$$= |\sum_t (P^O(S_t \neq s^D) - 0)\|\vec{c}\|_\infty| \tag{54}$$

$$\geq \sum_t (t\delta\kappa) \cdot \|\vec{c}\|_\infty \qquad \text{(Eq. (51) etc.)} \tag{55}$$

$$\geq \delta\kappa\|\vec{c}\|_\infty \sum_t t \tag{56}$$

$$\geq \delta\kappa'\|\vec{c}\|_\infty T^2 \tag{57}$$

for some $\kappa'$ (that absorbs both, $\kappa$ and the deviation between $T^2$ and $\sum_t t$.

*Proof for the lower bound, case $\delta T > 1$:*

First note that, alternatively to the previous deviation, we can also write $P^O(S_t \neq s^D)$ as the complementary probability of test-time constrained imitator *not* having deviated from the demonstrator state $s^D$ so far, i.e.,

$$P^O(S_t \neq s^D) = 1 - (1-\delta)^t \geq 1 - (1 - \frac{1}{T})^t, \tag{58}$$

since in the current case, $\delta T > 1$ and so $\delta > \frac{1}{T}$ and so $1 - \delta < 1 - \frac{1}{T}$ and so $(1-\delta)^t < (1 - \frac{1}{T})^t$ and so $1 - (1-\delta)^t > 1 - (1 - \frac{1}{T})^t$.

Now, for $t \geq \frac{T}{2}$, we get,

$$(1 - \frac{1}{T})^t \leq (1 - \frac{1}{T})^{\frac{T}{2}} = ((1 + \frac{-1}{T})^T)^{\frac{1}{2}} \leq (e^{-1})^{\frac{1}{2}} < 1, \tag{59}$$

where the second last inequality is Eq. (47). So $(1 - \frac{1}{T})^t$ is strictly below and bounded away from 1, and therefore Eq. (58) is uniformly strictly above and bounded away from 0, i.e., $P^C(S_t \neq s^D)$ is greater $\kappa'' > 0$ for $t \geq \frac{T}{2}$.

Then

$$|v^O - v^D| \tag{60}$$

$$= |\sum_{t=1}^{T} (P^O_{s_t} - P^D_{s_t}) \cdot \vec{c}| \tag{61}$$

$$= |\sum_{t=1}^{T} (P^O(S_t \neq s^D) - 0)\|\vec{c}\|_\infty| \tag{62}$$

$$\geq \sum_{t=\frac{T}{2}}^{T} P^O(S_t \neq s^D)\|\vec{c}\|_\infty \tag{63}$$

$$\geq \sum_{t=\frac{T}{2}}^{T} \kappa''\|\vec{c}\|_\infty \tag{64}$$

$$= \frac{T}{2}\kappa''\|\vec{c}\|_\infty \tag{65}$$

$$= \kappa'''T\|\vec{c}\|_\infty. \tag{66}$$

*Proof for the lower bound, final remarks:*

$\iota$ is given by taking the minima of the above $\kappa$'s and conbining it with the relation Eq. (46).

**Upper bound part:**

In what follows, $P$ stand for probability vectors/matrices, slightly overriding notation, and $P_{S'|S}$ for the transition probability (corresponding to $f$ plus potential noise).

First observe that generally (here $P$ stand for probability vectors/matrices, slightly overriding notation),

$$\Delta_{t+1} := \|P^O_{S_{t+1}} - P^D_{S_{t+1}}\|_1 \tag{67}$$

$$= \|P^O_{S'|S}P^O_{S_t} - P^D_{S'|S}P^D_{S_t}\|_1 \tag{68}$$

$$= \|P^O_{S'|S}P^O_{S_t} - P^O_{S'|S}P^D_{S_t} + P^O_{S'|S}P^D_{S_t} - P^D_{S'|S}P^D_{S_t}\|_1 \tag{69}$$

$$= \|P^O_{S'|S}(P^O_{S_t} - P^D_{S_t}) + (P^O_{S'|S} - P^D_{S'|S})P^D_{S_t}\|_1 \tag{70}$$

$$\leq \|P^O_{S'|S}(P^O_{S_t} - P^D_{S_t})\|_1 + \|(P^O_{S'|S} - P^D_{S'|S})P^D_{S_t}\|_1 \tag{71}$$

*First consider the first term in Eq. (71).* We can generally, just since the 1-norm of any stochastic matrix is 1, bound it as follows:

$$\|P^O_{S'|S}(P^O_{S_t} - P^D_{S_t})\|_1 \tag{72}$$

$$\leq \|P^O_{S'|S}\|_1 \Delta_t = 1\Delta_t \tag{73}$$

*Regarding the second term in Eq. (71),* we have the following way to bound it.

Let $\bar{S}$ denote the subset of the set $S$ of states where $\rho^D(s)$ has support, and, as in the main text, $\nu > 0$ is the minimum value of $\rho^D(s)$ on $\bar{S}$. Then

$$\nu \sum_{s \in \bar{S}} \|\pi^D(\cdot|s) - \pi^U(\cdot|s)\|_1 \tag{74}$$

$$= \nu \sum_{s \in \bar{S}, a} |\pi^D(a|s) - \pi^U(a|s)| \tag{75}$$

$$\leq \sum_{s \in \bar{S}, a} \rho^D(s)|\pi^D(a|s) - \pi^U(a|s)| \tag{76}$$

$$= \sum_{s,a} \rho^D(s)|\pi^D(a|s) - \pi^U(a|s)| \tag{77}$$

$$= \sum_{s,a} |\pi^D(a|s)\rho^D(s) - \pi^U(a|s)\rho^D(s)| \tag{78}$$

$$\tag{79}$$

$$= \sum_{s,a} |\pi^D(a|s)\rho^D(s) + \pi^U(a|s)\rho^U(s) - \pi^U(a|s)\rho^U(s) - \pi^U(a|s)\rho^D(s)| \tag{80}$$

$$\leq \sum_{s,a} |\pi^D(a|s)\rho^D(s) - \pi^U(a|s)\rho^U(s)| + |\pi^U(a|s)\rho^U(s) - \pi^U(a|s)\rho^D(s)| \quad \text{(triangle inequality)} \tag{81}$$

$$= \sum_{s,a} |\pi^D(a|s)\rho^D(s) - \pi^U(a|s)\rho^U(s)| + \pi^U(a|s)|\rho^U(s) - \rho^D(s)|? \tag{82}$$

$$\leq \sum_{s,a} |\pi^D(a|s)\rho^D(s) - \pi^U(a|s)\rho^U(s)| + |\rho^U(s) - \rho^D(s)| \tag{83}$$

$$= \|\rho^D_{s,a} - \rho^U_{s,a}\|_1 + \|\rho^D_s - \rho^U_s\|_1 \quad \text{(recall definition } \rho) \tag{84}$$

$$\leq 2\varepsilon + 2\varepsilon \quad \text{(by assumption and Eq. (37))}, \tag{85}$$

so in particular

$$\max_{s \in \bar{S}} \|\pi^D(\cdot|s) - \pi^U(\cdot|s)\|_1 \leq 4\varepsilon/\nu. \tag{86}$$

Therefore

$$\|P^D_{a|s|_{\bar{S}}} - P^U_{a|s|_{\bar{S}}}\|_1 = \|\pi^D_{a|s|_{\bar{S}}} - \pi^U_{a|s|_{\bar{S}}}\|_1 = \max_{s \in \bar{S}} \|\pi^D(\cdot|s) - \pi^U(\cdot|s)\|_1 \leq \frac{4\varepsilon}{\nu}, \tag{87}$$

where the first equation is just a rewriting (note that here, $\pi^D_{a|s|_{\bar{S}}}$ etc. denote matrices (transition matrix), while $\pi^D(\cdot|s)$ etc. denotes a vector (probability vector)). Note that this also implies the same bound for the test-time-constrained imitator, i.e.,

$$\|P^D_{a|s|_{\bar{S}}} - P^O_{a|s|_{\bar{S}}}\|_1 \leq \frac{4\varepsilon}{\nu}, \tag{88}$$

for the following reason: only the deviating mass between $D$ and $U$ that lies on actions where $D$ does not have support can be redistributed by the safety layer (because we assumed that $D$ is safe, and hence those actions are safe and are not affected by adding a safety layer, based on our assumption about the safety layer). But this deviation mass already lies where it has the maximum effect on the norm (namely: where $D$ does not have any mass at all). So redistributing it will make the deviation not bigger.

Furthermore we have, just by the definition of the conditional probability matrix $P_{S,A|S}$

$$\|(P^O_{S,A|S} - P^D_{S,A|S})\|_1 = \|(P^O_{A|S} - P^D_{A|S})\|_1, \tag{89}$$

and the same holds when restricting the state to any subset, in particular $\bar{S}$.

Now, to finalize our argument to also bound the second term in Eq. (71), observe

$$\|(P^O_{S'|S} - P^D_{S'|S})P^D_{S_t}\|_1 \tag{90}$$

$$= \|(P^O_{S'|S|_{\bar{S}}} - P^D_{S'|S|_{\bar{S}}})P^D_{S_t|_{\bar{S}}}\|_1 \quad \text{(since } \bar{S} \text{ is the demonstrator support)} \tag{91}$$

$$\leq \|(P^O_{S'|S|_{\bar{S}}} - P^D_{S'|S|_{\bar{S}}})\|_1 \|P^D_{S_t|_{\bar{S}}}\|_1 \tag{92}$$

$$= \|P_{S'|S,A}(P^O_{S,A|S|_{\bar{S}}} - P^D_{S,A|S|_{\bar{S}}})\|_1 \|P^D_{S_t|_{\bar{S}}}\|_1 \tag{93}$$

$$\leq \|P_{S'|S,A}\|_1 \|P^O_{S,A|S|_{\bar{S}}} - P^D_{S,A|S|_{\bar{S}}}\|_1 \|P^D_{S_t|_{\bar{S}}}\|_1 \tag{94}$$

$$= \|P_{S'|S,A}\|_1 \|P^O_{A|S|_{\bar{S}}} - P^D_{A|S|_{\bar{S}}}\|_1 \|P^D_{S_t|_{\bar{S}}}\|_1 \quad \text{(Eq. (89))} \tag{95}$$

$$\leq 1 \cdot \frac{4\varepsilon}{\nu} \cdot 1 \quad \text{(Eq. (88), 1-norm of stochastic matrix/vector is 1).} \tag{96}$$

Coming towards the end, observe that plugging Eq. (71), (73) and (96) together gives

$$\Delta_{t+1} - \Delta_t \leq \frac{4\varepsilon}{\nu}.$$

Therefore

$$\Delta_t = \sum_{t' \leq t} \Delta_{t'+1} - \Delta_{t'} \leq t\frac{4\varepsilon}{\nu}.$$

Then (keep in mind that we assumed that the reward depends only on the state)

$$|v^O - v^D| = |\sum_t (P^O_{s_t} - P^D_{s_t}) \cdot \vec{c}| \tag{97}$$

$$\leq \sum_t \|(P^O_{S_t} - P^D_{S_t})\|_1 \|\vec{c}\|_\infty \tag{98}$$

$$= \|\vec{c}\|_\infty \sum_t \Delta_t \tag{99}$$

$$\leq \|\vec{c}\|_\infty \sum_t t\frac{4\varepsilon}{\nu} \tag{100}$$

$$\leq \|\vec{c}\|_\infty \frac{4\varepsilon}{\nu} T^2. \tag{101}$$

$$\square$$

**Remark 4** (Remarks regarding Thm. 1 and proof)**.** *Some considerations:*

- *Why do we have this "min" lower bound in Eq. (7) of Thm. 1, and not just a purely quadratic one? Note that in the above proof, if the case $\delta > \frac{1}{T}$ holds, then $\delta T > 1$, but the maximum deviation between imitator and demonstrator cannot be larger than the imitator constantly being at a state different from the demonstartor with probability 1, i.e., can asymptotically not be larger than $T\|c\|_\infty$ (which is then smaller than $\delta T T \|c\|_\infty$ in this case).*

- *Note that the theorem formulation is about capturing all possible scenarios (within our overall setting), and thus also accounts for the "worst-case" ones. In practice, depending on the specific scenario, of course things may be "better behaved" for the test-time-only-safety approach than the (quadratic) error concluded by the theorem.*

## B   Additional background on GAIL cost and training

Overall we use (Wasserstein) GAIL, an established methodology, for imitation loss and training in our approach. The rough idea of this was already given in Sec. 4.2.

For the full account on details of (Wasserstein) GAIL, we refer the reader to the original work (Ho and Ermon, 2016; Xiao et al., 2019). Here let us nonetheless summarize some background on GAIL cost function and training, and comment on the parts relevant to our approach. (Additionally, some impelementation details on the version we use can be found in Sec. 5 and appendix C.)

### B.1 Imitation loss

Let us elaborate on the GAIL-based (Ho and Ermon, 2016) cost function. We do it more generally here to also include Wasserstein-GAIL/GAN.

The reason why we consider using the Wasserstein-version Xiao et al. (2019) is the following: Due to safety constraints, if the demonstrator is unsafe, we may deal with distributions of disjoint support. Then we want to encourage "geometric closeness" of distributions. But this is not possible with geometry-agnostic information-theoretic distance measures.

To specify the imitation cost $c$ that we left abstract in Sec. 2 and 4.2, let

$$c(s, a) = e_2(s, a), \tag{102}$$

$$(e_1, e_2) = \arg\max \mathbb{E}_{\pi^D}(\sum_t e_1(s_t, a_t)) + \mathbb{E}_{\pi^I}(\sum_t e_2(s_t, a_t)) \tag{103}$$

where the $\arg\max$ ranges over pairs $(e_1, e_2)$ of bounded functions from some underlying space of bounded functions (Xiao et al., 2019), for which, besides Eq. (103), the following is required (Xiao et al., 2019):

$$e_1(x) + e_2(y) \leq \delta(x, y), \tag{104}$$

for some underlying distance $\delta$. This condition is referred to as *Lipschitz regularity*. $e_1, e_2$ are called *Kantorovich potentials*.

As classic example, when taking $e_1(s, a) := \log(1 - D(s, a))$ and $e_2(s, a) := \log(D(s, a))$, for some function $D$, then this exactly amounts to the classic GAN/GAIL formulation with discriminator $D$.

### B.2 Remark on Training and the Handling of the State-Dependent Safe Action Set

Regarding the policy optimization step (c) sketched in Sec. 4.2, note that: For each sampled state $s_t$, $\pi^{I,\theta}(a|s_t)$ implicitly remembers the safe set $\tilde{A}_t^s$, such that also after any policy gradient step its mass will always remain within the safe set. Note that for this to work, it can be a problem if at some point we took a safe fallback action (Sec. 3.1) and thus only know a *singleton* safe set (and it is not a function of current state only). Since this is rare, it should pose little problems to training. Rigorously addressing this is left to future work.

### B.3 Comment on probability measure underlying the expectation

Observe that in Eq. (2) we take the expectation under the measure induced by dynamic system plus policy, over the summed future imitation cost. Alternatively, sometimes the expectation is taken under the time-averaged state-action distribution $\rho(s, a)$ (Sec. 3.3), over the imitation cost. As far as we see, both formulations seem to be equivalent, similar as is discussed in (Ho et al., 2016) for the discount-factor-based formulation.

## C   Additional details of experiments and implementation

Here we give some additional details for our experiments and implementation of Sec. 5. These details are not necessary for the rough idea, but to get the detailed picture of these parts.

Implementation code is available at: `https://github.com/boschresearch/fagil`.

### C.1 Additional architecture, computation and training information

Details of policy architectures:

- For the Gaussian policy (i.e., parameterizing mean and covariance matrix), we take two hidden layers each 1024 units.

- As pre-safe normalizing flow policy, we take 8 affine coupling layers with each coupling layer consisting of two hidden layers a 128 units.

- As discriminator, we take a neural net with two hidden layers and 128 units each.

- As state feature CNN, we take a ResNet18 applied to the abstract birds-eye-view image representation of the state (Chai et al., 2019).

For the custom neural nets, we use leaky ReLUs as non-linearities.

The data set (scene) consists of $\sim 1500$ tracks (trajectories), of which we use 300 as test set and 100 as validation, and the rest as training set.

One full safe set computation with some limited vectorized parts takes around 1.5s on a standard CPU. We believe vectorizing/parallelizing more of the code can further reduce this substantially.

For training, we use GAIL with training based on soft actor-critic (SAC) (Kostrikov et al., 2018; Haarnoja et al., 2018).

For the discriminator and its loss, we build on parameter clipping from Wasserstein GANs Arjovsky et al. (2017), and adversarial inverse reinforcement learning (AIRL) (Fu et al., 2017).

### C.2 Regarding safety guarantees and momentary safety cost

Regarding the fact that Prop. 2 does not strictly cover FAGIL-E yet: We conjecture that, extending Prop. 2, a guarantee can be given for the $l_\infty$ setting even without convexity over all dimensions, simply by the $l_\infty$ norm being a minimum over dimensions and then in some way considering each dimension individually.

Regarding Lipschitz continuity of momentary safety cost: Note that, as briefly stated in the main text, the momentary safety cost $d$ in our experiments is Lipschitz. Intuitively, it is some form of negative (minimum) *distance function* between ego and other vehicles. Distance/cost $d \leq 0$ is momentarily safe, while $d > 0$ is a collision. That is, $d$ is a Lipschitz continuous function, but $d = 0$ is the boundary between the set of safe states and unsafe states. (Similar to the safe action set being the sub-zero (i.e., $w \leq 0$) set of the total safety cost $w$.)

We believe that Lipschitz continuity is a fairly general assumption for safety, since safety is often considered in the physical world. And in the physical world, many of the underlying dependencies are in fact sufficiently continuous.

## D    Broader supplementary remarks

In this section, we gather several further discussions that are more broadly related to the results of the main part.

This section is not necessary to describe or understand the main part.

However, given that safe generative IL is a field that is rather little explored, we feel it can be particularly useful to provide some further comments here that may help understanding and may serve as a basis for next steps.

### D.1 Further details on Sec. 3.1

Several broader remarks regarding safe set inference, which are not necessary to understand the core definitions and results:

- A-temporally, Eq. (4) can be seen as a Stackelberg game.

- In Eq. (4), the others' maxima may often collapse to open-loop trajectories.

- We use the general formulation where policies $\pi_t$ can depend on $t$. This allows us to easily treat the case where we directly plan an *open*-loop action trajectory $a_{1:T}$ simultaneously, by simply $\pi_t(s_t) := a_t$, for all $t, s_t$.

- The state set corresponding to our action set would be an "invariant safe set" (Bansal et al., 2017).

- Lipschitz/continuity understanding is helpful also for other things: e.g., understanding the safe set's topology for Sec. 3.2.

- Our sample-based approach can also interpreted as follows: we can use *single-action safety checkers* (in the sense of the *"doer-checker"* approach (Koopman et al., 2019)) as oracles for action-set safety inference.

- On Prop. 2: Open-loop is a restriction but note that we do search over *all* trajectories in $\Pi, \Phi$, so it is similar to trajectory-based planning.

- *Regarding inner approximations of the safe set:* For as little as possible bias and conservativity, we need the inner approximation of the safe set to be as large as possible. In principle, one could also just give a finite set of points as inner approximation, but this could be very unflexible and biased, and injectivity for change-of-variables Prop. 3 would not be possible.

- *Regarding safety between discrete time steps:* Note that our setup is discrete time, but it can directly also imply safety guarantees between time stages: e.g., for our experiments, we know maximum velocity of the agents, so we know how much they can move at most between time stages, and we can add this as a margin to the distance-based safety cost.

- *Small set of candidate ego policies, to simplify calculations:* As an important simplification which preserves guarantees, in the definition/evaluation of safe set and total safety cost $w_t(s, a)$, we let the ego policies range over some small, sometimes finite set $\bar{\Pi}_{t+1:T} \subset \Pi_{t+1:T}$ of reasonable *fallback* future continuations of ego policies. In the case of autonomous driving (see experiments), $\bar{\Pi}_{t+1:T}$ consists of *emergency brake* and *evasive maneuvers* (additional details are in Appendix D.1).[21]

- A popular approach for trajectory planning/optimization is *model predictive control (MPC)* Bertsekas (2012), so let us briefly comment on the relation between our trajectory calculations for safety and MPC: In a sense, MPC could solve the trajectory optimization, i.e., the "min part" of Eq. (4). So, to some extent, it is related. However, MPC would need one specific policy/dynamics for the other agents, while we just have a set of possible other agents policies (and take the worst-case/adversarial case over this set). And MPC alone would also not give us a set of safe actions (which we need since we want to allow as much support for the generative policy density as possible), but only one action/trajectory.

- Note that the "adversarial" in the title of our work has the double meaning of "worst-case safety" and "using generative adversarial training".

---

[21] Regarding the ego, note that the search over a small set of future trajectories in the form of the *"fallback maneuvers"* (instead of an exhaustive search/optimization over the full planning/trajecotry space) can be seen as a simple form of motion primitives or graph search methods and related to "sampling-based" planning approaches (Paden et al., 2016). Regarding the other agents/perturbations, one could also just consider a small subset of possible behaviors, though we do not do this in the current work since it weakens guarantees of course. This would be somewhat related to sampling-based methods in continuous game theory (Adam et al., 2021) as well as empirical/oracle-based game theory (Lanctot et al., 2017).

## D.2 Further details on Sec. 3.2

Here we go more into detail on the construction of safety layer that are differentiable and give us an overall density, in particular (1) the challenges and (2) potential existing tools to build on. (See also Appendix D.3.2.)

One purpose of this section is to provide further relevant background for future work on safety layers that allow to have closed-form density/gradient – given this is a rather new yet highly non-trivial topic (from the mathematics as well as tractability side).

### D.2.1 Definition and remark

Let us introduce the topological notion of "connected" we use below.

Roughly, an open set in $\mathbb{R}^n$ is called *(path-) connected*, if any two of its elements can be connected by a continuous path within the set; and *simply connected* if these connecting paths are, up to continuous deformations within the set, unique.

For formal definitions we refer to textbooks, such as (Komornik, 2017).

Remark on Def. 1: Usually, parts $A_k$ will be topologically simply connected sets.

### D.2.2 Basic remarks on simplest safety layer approaches

Let us elaborate on the issue of designing safety layers, mappings $A \to \bar{A}$, in particular ones that are differentiable and give us an overall density.

Besides the more advanced approaches we discuss in the main text and below, let us also mention the two simplest ones and their limitations.

First, maybe the simplest safetly layer is to map all unsafe actions, i.e., $A \setminus \bar{A}$, to *one* safe action, i.e., *one point.* But this would lead to stronger biases and we would lose the property of having an overall (Lebesgue) density (at least the change of variables would not be applicable due to non-injectivity; and the policy gradient theorem would also be problematic).

Second, another simple solutions would be move/scale the whole $A$ into *one safe open set*, given the safe set contains at least one open set (which should generally be the case, e.g., using Prop. 1). This in fact would be a diffeomorphism so we could apply the change-of-variables formula. However, obviously this would be very rigid: the possible safe actions to take would then usually be a very small part of the full safe set – a strong and usually poor bias. This is because, be aware that, also the actually safe parts of the action space would be re-mapped to the tiny safe open set. That is, it would force us to also move actions that are actually safe to new positions. And this would leave little space of designing more sensible inductive biases, as in the two instances we propose.

### D.2.3 Simple versatility of piecewise diffeomorphism safety layers

In contrast to these overly biased simplest solutions from Appendix D.2.2, let us give an example of how piecewise diffeomorphisms give us more room for design. Note that he following is not meant as a safetly layer that should actually be used, but rather as a simple argument to show the universality:

**Remark 5.** *Piecewise diffeomorphic safety layers are versatile: If a given safe set $\bar{A}$ contains at least some area (open set), then we can usually map $A$ into $\bar{A}$, simply by moving and scaling the unsafe areas into safe ones, while leaving the safe actions where they are.*

*Elaboration of Rem. 5.* Assume the safe set $\bar{A}$ is given as the subzero set of some continuous function $h : A \to \mathbb{R}$, i.e., $\bar{A} = h^{-1}((-\infty, 0])$, which is the case, e.g., under Prop. 1, and that it contains at least some open set. Let $B$ be some (any) ball in this open set.

Here is the simplest construction: Just take $A_1 := h^{-1}((-\infty, 0])$ and $A_2 := h^{-1}((0, \infty))$. Let $g_1$ (on $A_1$) be the identity. And let $g_2$ (on $A_2$) be a shrinking, such that $A_2$ has less diameter than $B$, and a subsequent translation of $A_2$ into $B$.

Another construction, which has a better "topological" bias in that it tries to map unsafe parts into *nearby* safe parts works as follows:

If $\bar{A}$ is the subzero set of a continuous function, then the unsafe set is open. So it can be written as the union of its connected components, and each connected component is also open. Therefore, in each connected component $k$, there is at least one rational number (or, in higher dimensions, the analogous element of $\mathbb{Q}^n$), so there are at most countably many of them.

So define $g_k$ with domain the component $k$, and simply being a translation and scaling that moves this component $k$ into a *nearby* open part of the safe set (this is where it becomes more appealing in terms "topological" bias), or, as a default, into $B$.

Let $g$ be defined by the "union" of these $g_k$, and on the rest of $A$, i.e., on the safe set $\bar{A}$, let it simply be the identity.

$\square$

### D.2.4 On general existence, construction/biases and tractability of closed-form-density-preserving differentiable safety layer

Here we add some notes on the general discussion of safety layers with closed-form densities, beyond the particular approach we take in this paper.

There could be many ways to construct safety layers with closed-form density, and it is hard, especially within the scope of just this paper, to get general answers on questions of (1) possibility and (2) tractable construction of such safety layers. Especially since there may be trivial solutions (see Appendix D.2.2) which are not satisfactory in terms of rigidity/bias though. So let us narrow down the problem to try to make at least some basic assertions.

First, a *simple example of impossibility when using pure diffeomorphisms*: already for topological reasons, if $A$ is connected but $\tilde{A}$ not, then no such safety layer can exist that is bijective onto $\tilde{A}$.

Nonetheless, overall the starting point in this paper is the change of variables, or the piecewise change of variables we use. This implies that we extensively need diffeomorphisms to build our safety layer on. And this implies that we need homeomorphisms, when relaxing the differentiability to continuity.

Regarding the shapes of (un-)safe sets, or their (inner/outer) approximations, general answers would be desirable too, but particularly here we focus on polytopes and hyperrectangles, since for those are also typical results of reachability analysis etc. (Sidrane et al., 2022).

Here are some basic concepts and results from topology and related areas. We only give an idea, while detailed formalities are beyond the scope of this paper.

- In $\mathbb{R}^2$ there is a lot of relevant theory from complex analysis, which studies holomorphic, i.e., complex differentiable functions (another way to state it is that they are conformal, i.e., locally angle-preserving mappings; for detailed formalities see, e.g., (Luteberget, 2010)). Biholomorphic (i.e., a holomorphic bijection whose inverse is also holomorphic) implies diffeomorphic, so this is relevant for us.

  For instance, the *Riemann mapping theorem* (Luteberget, 2010) tells us that such a biholomorphic mapping exists from $A \subset \mathbb{R}^2$ to $\bar{A}$, *if $\bar{A}$ is a simply connected open set in $\mathbb{R}^2$* (assuming per se that $A$ is a box, i.e., simply connected as well). While this could be a powerful result for safety layers, the *issue comes with tractability.* Often, Riemann mappings are defined just implicitly, based on some variational problem, or some differential equation problem, whose solution can be arbitrarily complex to calculate. More tractable (though it still remains unclear how to make them fully tractable) concepts and results (Luteberget, 2010) include:

– Schwarz-Christoffel mappings – a special case for polytope domains;

– sphere packing-based algorithms for approximating Riemann mappings via combinatorical solutions on a finite graph of the problem, and then extrapolation to the continuous original problem.

– Note that it is easy to see (since holomorphic functions are globally uniquely determined from any local neighborhood, based on them being fully described by their Taylor expansion around any single point) that holomorphic functions are a quite rigid function class for safety layer though: for instance, there cannot be a holomorphic function which would be the identity on the safe set, but the non-indentity outside the safe set.

- Adding on the negative, i.e., non-existence side, often the non-existence of safety layers or parts of them can be proved using topological arguments (often invariants), e.g., that the domain and co-domain do have to have the same connectivity property (connected, simply conntected, etc.), or other based on homology.

- There are also positive, i.e., existence results, from topology, and very general ones. For instance, if two open subsets of $\mathbb{R}^n$ are contractible and simply connected at infinity, then they are homeomorphic (i.e., continuous bijections whose inverse is also continuous) to each other.[22] This, even though not sufficient, is a strong necessary requirement for being diffeomorphic of course. Also convexity can be helpful.[23]

### D.3   Details on FAGIL method Sec. 4, in particular its safety layers

### D.3.1   Regarding safe set inference module

Note that, in principle, due to modularity, many safety approaches can be plugged as safety modules into our overall method (Fig. 1). This includes frameworks based on set fixed points (Vidal et al., 2000), Lyapunov functions (Donti et al., 2021a), Hamilton-Jacobi type equations (continuous time) (Bansal et al., 2017), or Responsibility-Sensitive Safety (RSS) Shalev-Shwartz et al. (2017).

A fundamental problem in safe control is the trade-off between being safe (conservative) versus allowing for enough freedom to be able to "move at all" (actionability). Ours and many safe control frameworks build on more conservative adversarial/worst-case reasoning. RSS's idea is to prove safety under the less conservative (i.e., "less worst-case") assumption of others sticking to (traffic) *norms/conventions*, and if collisions happen nonetheless, then at least the ego is "not to blame".

### D.3.2   On the safety layers - an alternative, pre-safe-probability-based safety layer

Besides the safety layer instance we described in Sec. 4.1, which can be termed *distance-based*, here is another *pre-safe-probability-based* instance of a piecewise diffeomorphism safety layer $g$:

(a) On all safe boxes $k$, i.e., $A_k \subset \bar{A}$, $g_k$ is the identity, i.e., we just keep the pre-safe proposal $\hat{a}$. (b) On all unsafe boxes $k$, $g_k$ is the translation/scaling to the box where the pre-safe density $p_{\hat{a}}$ has the most mass (approximately). Alternatively, in (b) a probabilistic version (b') can be used where $g_k$ translates to safe part $A_\ell$ with (approximate) probability proportional to $p_{\hat{a}}$ on $A_\ell$.

Intuitively, the version with (b') amounts to a *conditioning on the safe action set*, i.e., a renormalization (when letting the box size go to zero).

Remark: Note that in this work we only considered these two specific instances of piecewise diffeomorphism safety layers – the pre-safe-probability-based and the distance-based one – while many others are imaginable.

---

[22] See https://math.stackexchange.com/q/55114/1060605.

[23] See https://math.stackexchange.com/q/165629/1060605 and http://relaunch.hcm.uni-bonn.de/fileadmin/geschke/papers/ConvexOpen.pdf.

### D.3.3    Illustrative examples of piecewise diffeomorphisms for safety layers

Let us give some illustrative examples of piecewise diffeomorphism safety layers, see Fig. 5. This includes an example of our *distance-based, grid-based safety layer* described in Sec. 4. (The probability-based, grid-based version from Appendix D.3.2 works in a related way, but choosing the targets $A_\ell$ based on their pre-safe probability, instead of their distance to the sources $A_k$.)

Note that in the grid-based example, here we just need translations, because all parts (rectangles) of the partition are of the same size, while in other cases, some parts (e.g., if there is a boundary not in line with the grid) may have different sizes, and then we also need scaling.

### D.3.4    Computational complexity of the safety layer

We here give a rough analysis of the computation complexity of piecewise diffeomorphisms (Sec. 3.2) and thus the safety layer, as part of the fail-safe imitator policy $\pi^{I,\theta}(\bar{a}|s)$ (Sec. 4), and their gradients. Let us look at Eq. (6) and recall the structure of the fail-safe imitator with safe action and pre-safe action from Fig. 1.

To calculate safe density $p_{\bar{a}}(\bar{a})$ of a safe action $\bar{a}$, given the pre-safe density $p_{\hat{a}}$ of the pre-safe policy, this is a sum (Eq. (6)) over the relevant parts $A_k$ of the partition $(A_k)_k$; and then each $k$-th summand is the change-of-variables formula applied to the diffeomorphism $g_k$ and pre-safe density $p_{\hat{a}}$, where the determination of $g_k$ may itself require some calculation (based, e.g., on distances or pre-safe densities of the parts).

So, essentially, the complexity is at most the number of parts in the partition $(A_k)_k$ times the complexity of calculating the change-of-variables formula (and determination of $g_k$ if necessary, see instances below).

Let us give some instances of this:

- When using any sort of general invertible (normalizing flow) neural net (Papamakarios et al., 2019) as transformations $g_k$, then we inherit their complexity for each summand of Eq. (6).

- For our proposed safety layers, e.g., the grid- and distance-based one we describe in Appendix D.3 the number of parts in the partition is around 100 (based on a 10 times 10 grid). And each $g_k$ is a translation/scaling to a target part, yielding a simple scaling factor as change-of-variables formula (the determination of the target part, i.e., of $g_k$, requires, for each state, one single run over all pairs of parts in the partition to get the distances; on this side, the probability-based safety layer version we describe in Sec. 4 is faster, since we do not have to go over all pairs, just all single ones).

To calculate the *gradient* of $\pi^{I,\theta}(\bar{a}|s)$ w.r.t. $\theta$, this is analogous, where we simply can drag the gradient into the sum of Eq. (6), and the gradients of the summands are given by the gradients of the pre-safe policy that is being used (i.e., for instance conditional Gaussian or normalizing flow), times the factor $|\det(J_{g_k^{-1}}(\hat{a}))|$ which is constant w.r.t. $\theta$ (in the our distance-based safety layer). So here the complexity is essentially number of (at most) all parts in the partition $(A_k)_k$, times complexity of gradient calculation for pre-safe policy.

### D.3.5    Invariant safety on full sets, not just singletons

Recall that for the general method we used singleton safe sets as backup to guarantee invariant safety (end of Sec. 4.1).

It is important to emphasize though that in *stable linear systems* it can in fact often be guaranteed that always a *full safe box $\bar{A}_k$ is found*. And then the extension of memorizing a safe fallback singleton is not necessary.

The idea towards proving this is as follows: The hard part is that we have to check safe future trajecotries not just for *individual* trajectories, but where at each stage, any action out of thes safe *set* can be taken. But these sets can be modelled as *bounded perturbations* of individual actions. Then stability arguments can show us that the error (deviation from a single safe trajectory over time) can be bounded, and then we can recur to planning individual trajectories plus simple margins.



**Setup:** Assume the full action set $A$ is given by the set enclosed by the gray frame. ...

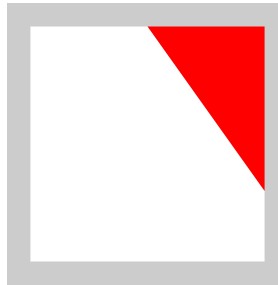

... And the unsafe actions $A \setminus \bar{A}$ are given by the red subset of the full action set.

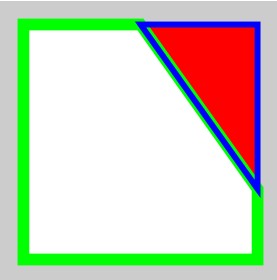

**First example piecewise diffeomorphism:** The partition $(A_k)_k$ has just two parts (l.h.s.): $A_1$ is the safe set enclosed by the green line, and $A_2$ the unsafe set, encircled by the blue line. The piecewise diffeomorphism $g$ transforms these parts as follows: ...

$\longmapsto$

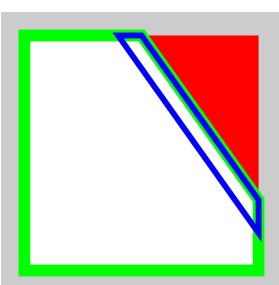

... The diffeomorphism $g_1$ on $A_1$ is just the identity. And the diffeomorphism $g_2$ maps the unsafe $A_2$ to the blue polygon at the boundary within the safe set (r.h.s.). Note that such a diffeomorphism exists by the Riemann (or Schwarz-Christoffel) mapping theorem, or, in this simple case, also by convexity (see our discussion of all these concepts in Appendix D.2.4). Together, $g_1$ and $g_2$ form our piecewise diffeomorphism $g$ on $A = A_1 \cup A_2$.

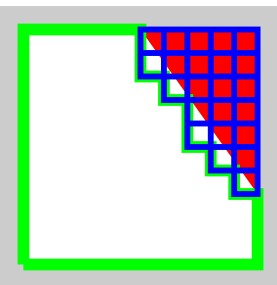

**Second example piecewise diffeomorphism:** Here, we consider our grid-based partition of the full action set, i.e., a complete regular tiling of the action set with rectangles $A_k$, as partition $(A_k)_k$. We only illustrated the individual rectangles $A_k$ for the unsafe area, in blue, while for the safe area we only drew the outer boundary, in green (l.h.s.). The piecewise diffeomorphism $g$ transforms these parts as follows: ...

$\longmapsto$

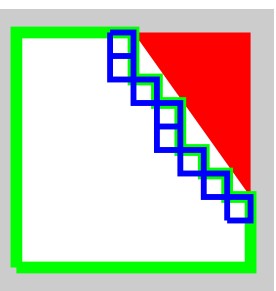

... On all safe $A_k$, the diffeomorphism $g_k$ is simply the identity. And on all unsafe $A_k$ (meaning fully or partially unsafe), $g_k$ is the shifting to the most nearby safe rectangle $A_\ell$. Those safe target rectangles $A_\ell$ are drawn in blue (r.h.s.). Together, these $g_k$ form our piecewise diffeomorphism $g$ on the full $A$. Note that this is an example of our *distance-based safety layer* described in Appendix D.3.2.

Figure 5: Examples of piecewise diffeomorphism safety layers. Top row: setup. Middle and bottom row: two examples.

### D.4 Further related work and remarks

Beyond the closest related work discussed in Sec. 1, let us here discuss some more broadly related work.

Boborzi et al. (2021) similar to us propose a safe version of GAIL, but there are several differences: they build on a different safety reasoning (RSS), which is specific for the scope of driving tasks, while our full method with its worst-case safety reasoning is applicable to general IL domains, and they do not provide exact densities/gradients of the full safe policy as we do.

Certain safety layers (Donti et al., 2021a; Dalal et al., 2018) are a special case of so-called *implicit layers* (Amos and Kolter, 2017; Amos et al., 2018; El Ghaoui et al., 2019; Bai et al., 2019; Ling et al., 2018; 2019), which have recently been applied also for game-theoretic multi-agent trajectory model learning (Geiger and Straehle, 2021), including highway driver modeling; robust learning updates in deep RL (Otto et al., 2021); and learning interventions that optimize certain equilibria in complex systems Besserve and Schölkopf (2021).

Worth mentioning is also work on safe IL that builds on hierarchical/hybrid approaches that combine imitation learning with classic safe planning/trajectory tracking (Chen et al., 2019; Huang et al., 2019).

Besides GAIL, which only requires samples of rollouts, one could also think about using fully known and differentiable dynamics (Baram et al., 2017) for the training of our method.

Also note reachability-based safe learning with Gaussian processes (Akametalu et al., 2014).

At the intersection of safety/reachability and learning, also work on outer approximations based on verified neural net function class properties is connected Sidrane et al. (2022).

Beyond the paper mentioned in the main text, there are several papers on fail-safe planning for autonomous driving (Magdici and Althoff, 2016; Pek and Althoff, 2018; 2020).

Convex sets, various kinds of them, have proved a helpful concept in reachability analysis and calculation, see., e.g, (Guernic and Girard, 2009) (their Section 1 gives an overview).

**Remark 6** (On the meaning of the term "fail-safe")**.** *The term "fail-safe" seems to be used in varying ways in the literature. In this work, in line with common uses, we have the following intuitive meaning in mind: We have an imitation learning component and a worst-case planning component in our system (the fail-safe imitator). The* pure IL component – *the pre-safe policy – can in principle always* fail *in the sense of leading to "failure", i.e., momentarily unsafe/collision, states, since it does not have strong guarantees.*[24] *Specifically, this would happen when it would put all mass on an action that leads to a collision.*[25] *But having our* worst-case fallback planner plus the safety layer *as additional component in our system, we make sure that, even if the IL component fails, nonetheless there exists at least one safe action plus future trajectory. This makes the system* safe against a failure of the pure IL component*, and this is what we mean by* fail-safe. *This is similar to (Magdici and Althoff, 2016) who use "fail-safe" in the sense that if other agents behave other than their "normal" prediction module predicts (i.e., the prediction module fails in this sense), then there is nonetheless a safe fallback plan. All this also relates to notions of* redundancy.

---

[24]There can also be some confusion because "fail" can be used in two senses: in the sense of a restricted failure, that can be mitigated and still be safe though; or in the sense that "failure" means "collision" or "momentarily unsafe" state.

[25]Note that for Gauss as well as flow policies, we have the conceptual issue here that they have mass everywhere; and in this probabilistic setting it is hard to say what precisely "fail" would be; however, one can as well plug in pre-safe policies into our approach that do not have support everywhere, and then the aforementioned holds rigorously.

