# OpenReview forum: "Fail-Safe Adversarial Generative Imitation Learning"
_TMLR — Accepted by TMLR_

### Review · Reviewer_Hc2K · 2022-08-19

**Summary Of Contributions:**

The paper proposes a method that aims to prevent unsafe actions during adversarial imitation learning by adding a differentiable safety layer,  which projects potentially unsafe actions to safe actions, at the end of the stochastic policy.
The method discretizes the action space into hyper-rectangles and tries to prove safety for each block under the assumptions of linear  dynamics and Lipschitz-continuity. More specifically, the paper proposes two methods: FAGIL-E evaluates at each corner, whether a best-case policy would encounter unsafe states, and FAGIL-L evaluates the safety-cost-to-go (negative for safe states) on the center actions and uses a Lipschitz argument. The safety-cost-to-go for given state-actions can be evaluated by solving a min-max problem that optimizes the remaining action sequence for worst-case dynamics.
If the input-action to the differentiable safety layer is in an unsafe block, a shift-and-scale transformation is performed to map it to the safe block that has the highest probability under the current policy. As shift-and-scale is a diffeomorphism, the probability density of the transformed action can be evaluated for learning the policy with GAIL.

The contribution of the submission are
1) A method for ensuring safe actions during imitation learning is proposed that works for low-dimensional, linear systems and convex safety sets.
2) A theorem is provided that the imitation cost scales linear with T when the safety layer is used during training and learning, and quadratic when it is only used during deployment.
3) The method is evaluated on a simple highway driving test problem (point mass, double integrator, two actions: one for accelerating in x, one for instantaneously switching the lane) and compared to  GAIL, RAIL (adding collision penalties to the learned cost function) and a variant that uses the safety layer only during deployment.

**Broader Impact Concerns:**

I do not think that the paper requires a broader impact statement.

**Requested Changes:**

* The feasibility of the method for problems with higher dimensional actions (e.g. 5-7) and nonlinear dynamics should be achieved / shown.

* Additional comparisons, e.g. with the method by Cheng et al. should be considered.

**Strengths And Weaknesses:**

* The claims seem correct

* The presentation is mostly clear, however, it should me explicit at some points. For example, how are shift-and-scale transformations implemented? Let's say we have a grid with safe set [0,1[ and unsafe set [1,2[. How would the shift & scale move the actions from [1,2] to [0,1[? It would seem reasonable to shift every action to the closest action 0.99, but we would'nt have a diffeomorphism. So should we transform [1,2] to, for example, [0.98, 0.99]? If I understand correctly the paper proposes to shift to whole [0,1[, but mapping to 0 seems unnecesary and suboptimal.

* The main weakness is that I dont see who would be interested in the findings of the paper, given that method makes extremely strong assumptions (very low-dimensional action space to allow for discretization, linear dynamics, convex safety sets). I dont see how the method or aspects of it (aside the general concept of using a safety layer) could be applied to a real-world application.

* The empirical evaluation is weak due to the simplicity of the considered environment. Furthermore, the work by Cheng et al. could be used as a baseline.


Cheng and Li Shen and Meng Fang and Liu Liu and Dacheng Tao, Lagrangian Generative Adversarial Imitation Learning with Safety

---

> ### Author Response · Authors · 2022-09-13
> **Response to Reviewer Hc2K I**
>
>
> We thank the reviewer for the detailed reading and feedback.
>
>
>
>
>
> > The presentation is mostly clear, however, it should me explicit at some points. For example, how are shift-and-scale transformations implemented? Let's say we have a grid with safe set [0,1[ and unsafe set [1,2[. How would the shift & scale move the actions from [1,2] to [0,1[? It would seem reasonable to shift every action to the closest action 0.99, but we would'nt have a diffeomorphism. So should we transform [1,2] to, for example, [0.98, 0.99]? If I understand correctly the paper proposes to shift to whole [0,1[, but mapping to 0 seems unnecesary and suboptimal.
>
>
> We added a graphical illustration of the piecewise diffeomorphism and safety layer, respectively, in Appendix D.3.3 of the revised paper version we uploaded.
>
>
> To respond to the specific example stated by the reviewer:
>
> - Yes, shifting all unsafe actions to, say, 0.99 (which is essentially what a non-generative projection safety layer like that of Donti et al. (2021a) would do) would not be a diffeomorphisms which is one of the key challenges.
>
> - In principle, in this simple 1-D case, of course once could have a simple piecewise diffeomorphism safety layer which, as the reviewers writes, shifts+scales [1,2] to something like [0.98, 0.99]. This (while keeping [0,1[ where it is) can be seen as a piecewise diffeomorphism with two "pieces" [0,1[ and [1,2[ as partition.
>
> - What we would do with our grid-based safety layer is as follows: Say the grid is 0.0, 0.1, 0.2, ..., 2.0. Then all line segmets (parts) [1.0, 1.1[, 1.1, 1.2[, ..., [1.9, 2.0[ would be shifted to [0.9, 1.0[. (Here we refer to the distance-based safety layer version described in D.3.2 which is most similar to your example; the other, probability-based one from Sec. 4, would choose the target part based on the probability of the pre-safe policy on that part instead of the distance).
>
>
>
>
>
>
> > Additional comparisons, e.g. with the method by Cheng et al. should be considered.
>
>
> The work by Cheng et al. on constrained/safe geneartive IL is clearly related to ours (though one difference is that they do not seem able to give safety guarantees beyond the training sample). Therefore we included it in the related work (Sec. 1) of the revised paper version.
>
> However, the one version of this paper which we found on the Web [https://openreview.net/forum?id=11PMuvv3tEO] is a ICLR 2022 submission that is marked as "withdrawn". Additionally, we could not find code for this method, neither in the paper nor otherwise on the Web. For us, this would speak against considering this as baseline.
>
> If the reviewer is aware of another version of the paper by Cheng et al., in particular publicly available code, we would, in principle, be willing to include this as an additional baseline in the paper.
>
> Overall, we understand the request of adding a further baseline and would be willing to do so.
>
> Another baseline candidate would be (Suo et al., 2021) or (Igl et al., 2022). However, also for thes baselines we could not find publicly available code. And these methods are highly non-trivial to implement, probably taking several weeks or even months to get them properly running (in particular since parts of these methods are just describen on a high level).
>
>
> Another possibility would be a baseline that one can implement oneself in an appropriate amount of time; right now, none comes to our mind, but we welcome proposals.

---

> > ### Author Response · Authors · 2022-09-13
> > **Response to Reviewer Hc2K II**
> >
> >
> >
> >
> >
> >
> > > The main weakness is that I dont see who would be interested in the findings of the paper, given that method makes extremely strong assumptions (very low-dimensional action space to allow for discretization, linear dynamics, convex safety sets). I dont see how the method or aspects of it (aside the general concept of using a safety layer) could be applied to a real-world application. ... The empirical evaluation is weak due to the simplicity of the considered environment. ... The feasibility of the method for problems with higher dimensional actions (e.g. 5-7) and nonlinear dynamics should be achieved / shown.
> >
> >
> > First, just so that everyone is on the same page, let us clarify: our *general-purpose theoretical safety results* in Sec. 3 do *not* require linearity, nor convexity, nor discretization*. We made this more clear in the revised paper version (Sec. 3.2 and conclusions). Specifically:
> >
> > - Prop. 1 just requires certain Lipschitz continuity conditions which are fairly general. Note that based on this Prop. 1, a much smarter sampling of the action safety cost function can be done than the simple grid/discretization. For instance using active learning approaches (e.g., Bryan et al.: "Active Learning For Identifying Function Threshold Boundaries"). However, this would probably need a dedicated paper, and, for the sake of the current paper's scope, we focused on a simpler implementation.
> >
> > - Prop. 3 just requires that *any* partition exists that allows for an appropriate piecewise diffeomorphisms. This requirement is fairly weak (we also elaborate on its generality in Rem. 5 in the Appendix). The grid/discretization based partition is just *one* possible such partition.
> >
> > - Note that Prop. 2 *does* require linearity and convexity, but is *not necessary* for our method FAGIL-L (whose safety is guaranteed by Prop. 1). We stated Prop. 2 as an alternative to Prop. 1 merely to show that also a different reasoning other than Lipschitz continuity can be used.
> >
> > - Note that Thm. 1 *does* use discrete spaces, but this result is not necessary for the safety guarantees of our method, it merely analyses the *imitation* advantage of methods like ours on a general level.

---

> > > ### Author Response · Authors · 2022-09-13
> > > **Response to Reviewer Hc2K III**
> > >
> > >
> > > Now, as the reviewer says, for the method instantiation we give for the low-dimensional case and use in the experiment, we *do* use a grid for the partition, and the double integrator transition. However, it needs to be emphasized:
> > >
> > > - The task of *driver imitation learning with low-dimensional, usually 2-D action space* has received *substantial interest from the autonomous driving community* in recent years (Bansal et al., 2018; Bhattacharyya et al., 2019; Suo et al., 2021)*, for control and simulation. The meaning/dynamics of the two dimensions can vary from ours, e.g., a non-linear bicycle model (Bhattacharyya et al., 2019). Note that, in this task, with its moving, uncertain obstacles (the other drivers), unrealistically/unreasonably *high collision rates in such driver imitation learning are generally considered a major open challenge* (Suo et al., 2021). General GAIL/RAIL have collision probabilities of above 10% (Table 1); Suo et al. have around 0.5% (Table 1 in Suo et al., 2021, of which no code was publicly available though). In contrast, we achieve 0.0% collisions, while coming close to imitation performance of GAIL.
> > >
> > > - It needs to be emphasized that *making guaranteed safety tractable is generally considered a key challenge in control* (Rungger and Tabuada, 2017; Gillula et al., 2014; Pek and Althoff, 2020; Fridovich-Keil and Tomlin, 2020). This holds even in *pure* control, i.e., even when not taking into account the additional challenge of *combining* safety with generative IL that we consider.
> > >
> > > - Therefore, for this paper, we made the trade-off of contributing, on the one hand, theoretical tools for general understanding and methodology towards safey generative IL; and, on the other hand, providing and evaluating tractable instantiations for the low-dimensional case (that covers, in particular, the driver imitation use case). This relates to common approaches for getting a hold on the overall problem, such as concentrating on linear settings (Rungger and Tabuada, 2017; Gillula et al., 2014), and/or using various forms of inner/outer approximations, including separation of dimensions (Pek and Althoff, 2020; Gillula et al., 2014). One future step to mitigate the curse of dimensionality may lie in smarter sampling of the action space via active learning, as mentioned above.
> > >
> > > - Regarding linearity: while we use the 2-D double integrator as transition, note that the overall dynamics of our experimental setting is not linear (the overall dynamics includes things such as the non-linear shortest-time controllers als fallback maneuvers). Intuitively, our low-dimensional instantiation does not so much rely on linearity (the Lipschitz argument from Prop. 1 also works for non-linear dynamics), but rather on *separability* of dimensions, which is one common way to simplify or approximate things (Pek and Althoff, 2020). As (Pek and Althoff, 2020) does, such separations can also be used as provable inner/outer approximations for non-linear dynamics (which was beyond the scope of our paper). We made this more clear in Sec. 5 and 6 of the revised paper version we uploaded.
> > >
> > > - (As a minor point on our low-dimensional setup, note that the lateral vehicle motion is modelled by a *continuous* acceleration action analogous to the longitudinal.)
> > >
> > >
> > > This being said, of course *we agree with the reviewer* that our implementation's focus on the low-dimensional, grid-based case *does* consitute a *limitation of our work*. And therefore we made this limitation more clear in the limitations paragraph in the conclusion section of the revised paper version.
> > >
> > > And, as we mentioned, we would be willing to include another appropriate baseline if the code is publicly available (or if it is easy enough to implement oneself).
> > >
> > >
> > > Overall, we hope that the reviewer is willing to consider a technically correct paper, that combines a focus on general theoretical results with an experimental evaluation in a restricted but application-oriented setting,
> > > as one relevant self-contained research step towards safe/robust IL.

---

### Review · Reviewer_d5hT · 2022-08-26

**Summary Of Contributions:**

In this work the authors propose a method for safe imitation learning which is numerically tested on real-world data.

**Broader Impact Concerns:**

None.

**Requested Changes:**

I think the paper needs to be completely rewritten with a focus on clarity. As it currently stand, it is just too confusing to properly evaluate. Maybe it is just me, but I have read the manuscript multiple times and have yet to understand much of what the authors are trying to convey.

**Strengths And Weaknesses:**

Strengths.

The theory in the paper appears to be correct and the method proposed by the authors seems to perform well.

Weaknesses.

However, I have a major concern. Overall I found the paper to be written in an extremely confusing manner. Take, for example, the abstract of the paper.

(i) [...] The safety layer is a “glueing together” of piecewise diffeomorphisms, with sum over change-of-variables formulas as density. [...] means nothing to the reader without having read the paper previously.

(ii) [...] The safe action set (into which the safety layer maps) is inferred by sample-based adversarial reachability analysis of fallback maneuvers plus Lipschitz continuity or convexity arguments. [...] neither does this.

A lot of comments are thrown around throughout the manuscript with little context or information to the reader. This makes it very confusing to understand what type of insight about their method the authors are trying to provide.

Furthermore, the paper has an a lot of footnotes (24 in total), two remarks and a 6-page Appendix B titled 'Further Details'. If that many clarifications are needed, the paper should be better organized and explained.

---

> ### Author Response · Authors · 2022-09-13
> **Response to Reviewer d5hT I**
>
> We thank the reviewer for the feedback.
>
>
>
> > (i) [...] The safety layer is a “glueing together” of piecewise diffeomorphisms, with sum over change-of-variables formulas as density. [...] means nothing to the reader without having read the paper previously.
>
> We understand the reviewer's point, but also consider it important to mention the most important technical terms already in the abstract for those readers that are already familiar with these terms.
>
> Therefore we take the middle ground, by keeping the key technical terms but also adding more intuitive parts. Here is the modified sentence of the revised version we uploaded:
>
> "The safety layer maps unsafe actions into a set of safe actions, and uses the change-of-variables formula plus additivity of measures for the density."
>
>
>
> > (ii) [...] The safe action set (into which the safety layer maps) is inferred by sample-based adversarial reachability analysis of fallback maneuvers plus Lipschitz continuity or convexity arguments. [...] neither does this.
>
>
> Analogous to the above, we modify this sentence in the revised manuscript version to:
>
> "The set of safe actions is inferred by first checking safety of a finite sample of actions via adversarial reachability analysis of fallback maneuvers, and then concluding on the safety of these actions' infinite neighborhoods using, e.g., Lipschitz continuity."

---

> > ### Author Response · Authors · 2022-09-13
> > **Response to Reviewer d5hT II**
> >
> > > A lot of comments are thrown around throughout the manuscript with little context or information to the reader. This makes it very confusing to understand ... the paper has an a lot of footnotes (24 in total), two remarks and a 6-page Appendix B titled 'Further Details'. ... I think the paper needs to be completely rewritten with a focus on clarity.
> >
> >
> > One unclearity we read from the reviewer's feedback is which *role the appendix plays* in relation to the parts in the main paper (where the appendix is referred to).
> >
> > Therefore, in the revised manuscript version, we clarified this role at all relevant places in main part and appendix:
> >
> > - by explicitly describing the role/relevance of the appendix at the respective places where we refer to it: E.g., in Sec 3.1 we write: "For several broader remarks on the results in this section, which are not necessary to understand this main part though, see also Appendix D.1.". Sec 3.2: "For further remarks on motivation and details of piecewise diffeomorphism layers, which are not necessary to understand this main part though, we refer the interested reader to Appendix D.1. And similarly in Sec 4.2
> > - Also in Sec. 3.2 we simplified the example such that topology can be avoided (and moved the topological example to the appendix).
> > - In the appendix, we split up the previous section B ("Further Details"), and now make explicit at the beginning of each section their respective role.
> >
> > While we understood these points and happily modified them in the revised version accordingly,
> > we do not understand the fundamental critizism of clarity:
> > - We spent a lot of time on structuring and writing the paper, providing motivation and context (in the introduction and varius other places); intuitions (figures 1-3); carefully writing and checking all proofs (Appendix 1), based on proper introduction of all needed definitions and concepts (mainly Sec. 2; or explicit reference to related work if not written explicitly).
> > - This paper necessarily requires to use concepts from different areas, including generative IL, change-of-variables formula, and reachability analysis. We hope the reviewer sees that such a paper requires a trade-off of how long and explicit each of the areas can be introduced. And while *of course all used areas do have to be properly introduced in a paper*, sometimes such introduction has to be short, and interested readers are referred to references or appendix. This is in order to keep the exposition interesting and concise also for those readers, who are in fact already familiar with some or all areas we build on.
> >
> >
> > Specifically,
> > - regarding footnotes: we have 24 footnotes (including appendix) on more than 30 pages (including appendix). We do not think this is excessive. And generally we do not see the problem with using footnotes. Instead we view footnotes as a common and helpful stylistic instrument (essentially, if the reader is fine with what he/she reads in the main text part (understands it well enough), then he/she does not have to look at the footnote and can just continue the read flow; however, if, at the specific point of the footnote, the reader is interested or feels that he/she needs more info, then he/she reads the footnote.) For one particular footnote -- number 4 -- we did however see now that it was quite long and shortened it.
> > - Regarding remarks: as we feel is common, we use the "Remark" environments (Remark 1 and 2 in the main text) as stylistic instrument to highlight parts that are particularly relevant, and to which one wants to refer to at other places in the text.
> >
> >
> > Overall, we hope that the modifications we made in the revised version improved the clarity w.r.t. the points raised by the reviewer.
> >
> > We are willing to make further modifications, but we ask the reviewer to take into account, that for this we need concrete input on what specifically the reviewer sees as problematic.
> >
> > Because we do not see right now what else can be done to improve clarity.

---

### Review · Reviewer_WeWp · 2022-08-30

**Summary Of Contributions:**

The authors propose a novel safe imitation learning algorithm by adding a safety layer on the actions to ensure that only safe actions are taken. The proposed approach uses GAIL-style training, along with a differentiable safety layer to train end-to-end. Results show that using the safety layer during training is better than only using it at test time. In a simulated driving environment, the proposed approach reaches good imitation learning performance while avoiding collisions.

**Broader Impact Concerns:**

It would be good to have more discussion about broader impacts. In particular, this paper focuses on autonomous driving and safety, so I would like to see some discussion of negative and positive effects of fielding systems like the one proposed in this paper.

**Requested Changes:**

Why is the total safety cost to go in Eq (4) not a sum over time. It appears to currently just pick out the least safe state in the future rather than a total cost. (critical)

Prop 1: Is the momentary safety cost d Lipschitz? It seems like in driving and other safety-critical domains a very slight change in state can be the difference between safety or collision. Please clarify and discuss when this assumption is likely to hold and when it will not (strengthen).

I found the section on piecewise diffeomorphisms confusing. Is there a way to provide a visual example to help readers gain intuition? (strenthen)

Also, the implied density of y seems quite complex and compute intensive to calculate the gradient of. Can you provide more details on the complexity and scalability of the proposed approach? (critical)

Missing important related work on safe and robust imitation learning and shielding for RL (critical):
- Javed, Zaynah, et al. "Policy gradient bayesian robust optimization for imitation learning." International Conference on Machine Learning. PMLR, 2021.
- Alshiekh, Mohammed, et al. "Safe reinforcement learning via shielding." Proceedings of the AAAI Conference on Artificial Intelligence. Vol. 32. No. 1. 2018.
- Brown, Daniel, et al. "Safe imitation learning via fast bayesian reward inference from preferences." International Conference on Machine Learning. PMLR, 2020.
- Lacotte, Jonathan, et al. "Risk-sensitive generative adversarial imitation learning." The 22nd International Conference on Artificial Intelligence and Statistics. PMLR, 2019.
- Thananjeyan, Brijen, et al. "Recovery rl: Safe reinforcement learning with learned recovery zones." IEEE Robotics and Automation Letters 6.3 (2021): 4915-4922.

Open loop simulation means the agents don't react to changes in ego agent. Seems unrealistic, but necessary if just replaying observational driving data. Is there a different simulation or experiment that could be added to the paper? (strengthen)

If you can check the safety of actions, this seems to require a model of other drivers and the road, so why not just run MPC? Please clarify (critical)

In particular, I'm confused about how the total safety cost is computed in practice. It seems like a difficult optimization problem and it's unclear to me how this is done in the driving simulation and how generally it could be applied. (critical)

The safety fallback is confusing. It's unclear to me how you can fall back on a safe policy if there is no fully safe part of A_k. Doesn't that mean that there are no safe actions. This seems to contradict the fact that there is a fall back policy you can use. Please clarify (critical)

Please add a limitations section. Can you highlight failure modes of the proposed system? Maybe show what happens when the assumptions are violated? (critical)

**Strengths And Weaknesses:**

Strengths
- The approach seems novel and interesting from a theoretical perspective
- Empirical results show that the approach performs well compared to alternatives
- Good theory and visual example in Fig 2 of why the safety layer is important for training.

Weaknesses
- Many details are confusing and it is unclear how well this approach will scale to more complex problems
- Missing some related work on safe IL and safe RL.
- Unclear what the limitations of this approach are and when it should be used and when it is inappropriate.

---

> ### Author Response · Authors · 2022-09-13
> **Response to Reviewer WeWp I**
>
> We thank the reviewer for the detailed reading and feedback. Below (and in the revised manuscript version), we address all the reviewer's points except for some of those marked by "strengthen" (for time reasons), but would be happy to also include the latter in the final version.
>
>
>
>
>
> > Why is the total safety cost to go in Eq (4) not a sum over time. It appears to currently just pick out the least safe state in the future rather than a total cost. (critical)
>
> To explain this, we added the following sentence in Sec. 3.1 of the revised version:
>
> "Note that requiring momentary safety cost $d(s_{t'}) \leq 0$ *for each* future $t'$ in Eq. (3) translates to requiring that  $d(s_{t'}) \leq 0$ for the *maximum* over future $t'$ in Eq. (4)", where Eq. (4) defines the total safety cost. Similar formulations can be found, e.g., in (Fridovich-Keil and Tomlin, 2020).
>
>
> > Prop 1: Is the momentary safety cost d Lipschitz? It seems like in driving and other safety-critical domains a very slight change in state can be the difference between safety or collision. Please clarify and discuss when this assumption is likely to hold and when it will not (strengthen).
>
>
> Yes, the momentary safety cost $d$ in our experiments is Lipschitz. Intuitively, it is some form of *negative* minimum *distance* between ego and other vehicles. Distance/cost $d < 0$ is safe, while $d > 0$ is a collision.
>
> That is, $d$ is a Lipschitz continuous function, but $d=0$ is the boundary between the set of safe states and unsafe states. (Similar to the safe action set being the sub-zero (i.e., $w < 0$) set of the total safety cost $w$ (Sec. 3.1).)
>
> We believe that Lipschitz continuity is a fairly general assumption for safety, since safety is often considered within the realm of the physical world; and in the physical world, many of the underlying dependencies are in fact sufficiently continuous.
>
> We added this in Appendix C.2 of the revised version.
>
>
>
> > I found the section on piecewise diffeomorphisms confusing. Is there a way to provide a visual example to help readers gain intuition? (strenthen)
>
>
> We added several figures with examples of piecewise diffeomorphisms / safety layers in Appendix D.3.3 of the uploaded revised version.
>
>
>
>
> > Also, the implied density of y seems quite complex and compute intensive to calculate the gradient of. Can you provide more details on the complexity and scalability of the proposed approach? (critical)
>
> We added a description of the compute complexity of the piecewise diffeomorphisms as safety layers in the fail-safe imitator policy, and its gradient, in Appendix D.3.4 of the revised manuscript version.
>
> Essentially, by looking at Eq. (5) of the paper, the complexity of the overall density is the sum with at most the number of parts $A_k$ many summands, and each summand requires computation of the determinant of $g_k$ plus computation of the pre-safe density (i.e., computation of pre-safe policy density, like conditional Gauss or flow). The number of parts in our grid-based implementation for the experiment is around 10*10 = 100. The $g_k$ can, e.g., be a simple scaling/translation, which is used in ours grid-based instantiation, and then the determinant is just the scaling factor; or it can be as complex as a neural net normalizing flow, and then the computation complexity is given by the latter. The gradient calculation is similar, where we can drag the gradient into the sum, and then for each summand, the gradient calculation is dominated by the gradient calculation of the pre-safe policy.
>
> So, roughly speaking, the computation complexity of density/gradient is *[number of parts in the partition] times [complexity of $g_k$ (or gradient of pre-safe policy)]*.
>
>
>
>
> > Missing important related work on safe and robust imitation learning and shielding for RL (critical): ...
>
> Thanks, we added all these references to our related work (Sec. 1).
>
>
>
>
> > If you can check the safety of actions, this seems to require a model of other drivers and the road, so why not just run MPC? Please clarify (critical)
>
>
> What we assume to know is the transition function and the set of possible behavior by the other agents and/or noise (Sec. 2). What we need to calculate is the safe action set, or, equivalently, the sub-zero set of the total safety cost (Sec. 3.1).
>
> In a sense, MPC could solve the trajectory optimization, i.e., the "min part" of Eq. (4). So, to some extent, it is related.
>
> However, MPC would need one specific policy/dynamics for the other agents, while we just have a set of possible other agents policies (and take the worst-case/adversarial case over this set). And MPC alone would also not give us a set of safe actions (which we need since we want to allow as much support for the generative policy density as possible), but only one action/trajectory.
>
> We added this comment in Section D.1.

---

> > ### Author Response · Authors · 2022-09-13
> > **Response to Reviewer WeWp II**
> >
> > > In particular, I'm confused about how the total safety cost is computed in practice. It seems like a difficult optimization problem and it's unclear to me how this is done in the driving simulation and how generally it could be applied. (critical)
> >
> >
> > In general, the computation of safe sets and safety costs is in deed a non-trivial problem, and so far there clearly is no tractable one-fits-all solution. Instead there are just specific tractable methods that have been found for restricted settings and/or using forms of over-/under-approximations (Rungger and Tabuada, 2017; Gillula et al., 2014; Fridovich-Keil and Tomlin, 2020; Pek and Althoff, 2020). And essentially, many of those approaches can be "plugged" into our approach.
> >
> > Specifically, for the method instantiation used in the experiments, the calculation of total safety cost $w$ and safe action set is described in Sec. 5, and we slightly expand this description in the revised version:
> >
> > "*Regarding safe set inference:*  In this setting, *where the two action/state dimensions and their dynamics are separable*, the others' reachable rectangles can simply be computed exactly based on the others' maximum longitudinal/lateral acceleration/velocity (this can be extended to approximate separability (Pek and Althoff, 2020)). As ego's fallback maneuver candidates, we use non-linear shortest-time controllers to roll out emergency brake and evasive maneuver trajectories (Pek and Althoff, 2020). Then the total safety cost $w$ is calculated by taking the minimum momentary safety cost $d$ (as defined above, i.e., minimum distance) between ego's maneuvers and others' reachable rectangles over time. Note that the safety cost $d$ and dynamics with these fallbacks are each Lipschitz continuous as required by Prop. 1, implying the *safety guarantee* for FAGIL-L."
> >
> >
> >
> >
> >
> > > The safety fallback is confusing. It's unclear to me how you can fall back on a safe policy if there is no fully safe part of A_k. Doesn't that mean that there are no safe actions. This seems to contradict the fact that there is a fall back policy you can use. Please clarify (critical)
> >
> > The parts $A_k$ are sets that usually consist of more than one action (in our experimental instance, the $A_k$ are rectangles). So it may happen that not the *full set* $A_k$ is safe, but nonetheless one or some actions *in* this set $A_k$ are safe.
> >
> > We elaborated the explanation of this in Sec. 4.1 of the uploaded revised version.
> >
> >
> >
> >
> > > Please add a limitations section. Can you highlight failure modes of the proposed system? Maybe show what happens when the assumptions are violated? (critical)
> >
> > We see the relevance of making limitations explicit. Therefore we rewrote the conculsions, adding a detailed limitations paragraph (we feel this fits better than an entire section):
> >
> > "In this paper, we considered the problem of safe and robust generative imitation learning ...
> >
> > We contributed to filling several gaps that existed so far towards solving this problem:
> > inference of guaranteed adversarially safe action sets,
> > safety layers with closed-form density/gradient,
> > and the theoretical understanding of end-to-end generative training with safety layers.
> > Combining these results, we described a general abstract method for the problem,
> > as well as one specific tractable instantiation for a restricted low-dimensional setting. ...
> >
> > Limitations of our work come from the core challenges that safety goals often bring along in terms of tractability and cautiousness:
> > First, the grid-based instantiation we give of our general theory is tractable mainly in the low-dimensional case, and in the experimental setting we harnessed tractability via separability of longitudinal and lateral dynamics of the other agents.
> > Second, worst-case safety can lead to overly cautious actions while human ``demonstrators'' often achieve surprisingly good trade-offs in this regard; incorporating more restrictions about other agents, such as the responsibility-sensitive safety (RSS) framework, would still be covered by our general theory but help to be less conservative.
> > Additionally, in the experimental evaluation, we concentrated on one challenging but restricted setting,
> > leaving an extensive purely empirical study to future work."
> >
> > > It would be good to have more discussion about broader impacts. ...
> >
> >
> > We added a broader impact statement in the revised version (after conclusions).

---

### Author Response · Authors · 2022-09-13
**Introductory Summary Review Response and Revised Manuscript**

We would like to thank all reviewers for their feedback.

Here we first summarize the main aspects we see. Details follow below in our individual responses, and the revised manuscript version we uploaded.

Regarding *technical soundness*, to all reviewers the *theoretical claims appear "correct" (d5hT, Hc2K) or "good"* (WeWp, regarding Thm. 1). We are happy to read this, since the theoretical safety guarantees and analysis of safety layers for IL are at the core of our contribution.

Regarding *the experimental evaluation* of our approach on real-world driver data, *two reviewers (WeWp, d5hT) consider it to perform well*.

Regarding *clarity*: while reviewer Hc2K considers the presentation as "mostly clear", WeWp lists specific confusing details, and d5hT criticizes the overall clarity. While we believe that already the submitted version was mature, we are happy about the constructive feedback. Accordingly, we made modifications in the revised version for all concrete criticisms we could take from the reviews (for WeWp, we focused on those marked as "critical"), in particular, added clarifications. Here we list all modifications while details follow in the individual responses:
- [WeWp & Hc2K] Appendix D.3.3: illustration/explication of piecewise diffeomorphism safety layers
- [WeWp] Sec. 1: added related work
- [WeWp] Sec. 3.1, Sec. 4.1, Sec. 5, Sec D.1, Appendix D.3.4: clarified the respective details
- [WeWp & Hc2K] Sec. 6: added paragraph with discussion of limitations to conclusions
- [WeWp] added Broader Impact Statement
- [d5hT] Abstract: more informal explanations that require less technical terminology
- [d5hT] Sec 3.1, Sec 3.2, Sec 4.2: simplified example, clarified role of Appendix when referring to it, among other things
- [d5hT] Appendix: split up, restructured, and contextualized the previous long Appendix B
- [Hc2K] Sec. 1: added related work Cheng et al.
- [Hc2K] Sec. 3.2, Sec. 5: comments and elaborations on assumptions

Beyond this, we do not see right now what else can be done to improve clarity (except for the remaining "strengthen" points of reviewer WeWp). Of course we would be willing to make further modifications for clarity, but we hope that it is understandable, that in order to do this, we need *concrete* input on what specifically is seen as unclear.

Regarding *limitations*,
reviewer WeWp asks for a discussion of limitations,
and reviewer Hc2K considers the strength of assumptions and the implications on relevance as main weakness. While we provide elaborated responses below, let us here emphasize, just so that everyone is on the same page:
- Our *general-purpose theoretical safety results* in Sec. 3 do *not* require linearity, nor convexity, nor discretization. Therefore, our abstract general method (Sec. 4) that combines these results, does not require these assumptions. However, for the specific method instantiation we give and evaluate in the experiment, we do rely on a grid and therefore focus on the low-dimensional case. We made this more clear in the revised manuscript version. (For linearity aspects of the experiments see below.)
- The task of *driver imitation learning with low-dimensional, usually 2-D action space* has received *substantial interest from the autonomous driving community* in recent years (Bansal et al., 2018; Bhattacharyya et al., 2019; Suo et al., 2021; Igl et al., 2022), for control and simulation.
- *Making guaranteed safety tractable is generally considered a key challenge in control* (Rungger and Tabuada, 2017; Gillula et al., 2014; Pek and Althoff, 2020; Fridovich-Keil and Tomlin, 2020). This holds even in *pure* control, i.e., without the combination with IL.

This being said, we do see that our implementation's and experiment's focus on the low-dimensional case constitutes a main limitation of our work; and that it is important to *make such limitations explicit*. Therefore we include a detailed limitations paragraph in the conclusions of the revised version.

Beyond that, while we believe RAIL, GAIL and TTOS are natural baselines (Table 1) for our approach, we would be willing to include a further baseline (that can be implemented in a reasonable amount of time; for the candidate Cheng et al. we elaborate on the issues in the response to Hc2K).

---

### Decision · Action_Editors · 2022-10-03

**Recommendation:** Accept as is

**Comment:**

This paper deals with safe imitation learning, through safety layers. The authors claim that training the imitative agent with a safety layer is more efficient than adding the safety layer only at test time, which is supported by both theoretical (Rk 1 and Thm1; linear in T upper-bound vs quadratic in T lower-bound, roughly) and empirical (on safe autonomous driving) evidences. The safety layer is a hand-crafted part of the policy network, and should be differentiable for being considered at train time. For designing this layer, the authors justify checking the safety of a finite sample of actions using eg a Lipschitz argument (Prop 1), and use piecewise diffeomorphisms (Prop 3) for both differentiability and ease of design.

Overall, the reviewers found the approach to be technically sound and performing well on the proposed experiment. Yet, they raised concerns about the clarity of the initial submission and its significance (notably due to the lack of scalability of the specific instantiation of the safety layer considered in this paper). The authors proposed a revision which improves the clarity and discusses/addresses the limitations in a better way (notably by making them clearer).

After rebuttal, revision and discussion, all reviewers lean toward accepting the paper, I thus recommend acceptance. The paper does have limitations (eg, the practical safety layer considered here might not be very scalable), but they are properly discussed, and the authors make a good point in advocating for considering the safety layer already at train time for imitation learning, both from theoretical and empirical viewpoints. The considered empirical setting (safe autonomous driving), even if of low dimension, is not obvious and is of practical interest.

**Audience:**

Safe imitation learning, and the contributions of this paper, should be of interest for part of the TMLR's audience.

**Claims And Evidence:**

The claims made in this submission are supported by both theoretical and empirical evidences.

---

> ### Author Response · Authors · 2022-11-05
> **Thanks to action editors and reviewers**
>
> Dear action editors and reviewers,
>
> Now we uploaded the camera ready version.
>
> Thanks a lot again to the reviewers for their feedback.
>
> Thanks a lot also to the action editors for their detailed remarks.
>
> Best,
> the authors